# Age and pattern of the southern high-latitude continental end-Permian extinction constrained by multiproxy analysis

Christopher R. Fielding [1], Tracy D. Frank [1], Stephen McLoughlin [2], Vivi Vajda [2], Chris Mays[2], Allen P. Tevyaw[1], Arne Winguth[3], Cornelia Winguth[3], Robert S. Nicoll[4], Malcolm Bocking[5] & James L. Crowley[6]

Past studies of the end-Permian extinction (EPE), the largest biotic crisis of the Phanerozoic, have not resolved the timing of events in southern high-latitudes. Here we use palynology coupled with high-precision CA-ID-TIMS dating of euhedral zircons from continental sequences of the Sydney Basin, Australia, to show that the collapse of the austral Permian *Glossopteris* flora occurred prior to 252.3 Ma (~370 kyrs before the main marine extinction). Weathering proxies indicate that floristic changes occurred during a brief climate perturbation in a regional alluvial landscape that otherwise experienced insubstantial change in fluvial style, insignificant reorganization of the depositional surface, and no abrupt aridification. Palaeoclimate modelling suggests a moderate shift to warmer summer temperatures and amplified seasonality in temperature across the EPE, and warmer and wetter conditions for all seasons into the Early Triassic. The terrestrial EPE and a succeeding peak in Ni concentration in the Sydney Basin correlate, respectively, to the onset of the primary extrusive and intrusive phases of the Siberian Traps Large Igneous Province.

[1] Department of Earth & Atmospheric Sciences, University of Nebraska-Lincoln, 126 Bessey Hall, Lincoln, NE 68588-0340, USA. [2] Swedish Museum of Natural History, Box 50007 S-104 05 Stockholm, Sweden. [3] Department of Earth & Environmental Sciences, University of Texas at Arlington, PO Box 19049 Arlington, TX 76019, USA. [4] Geoscience Australia, GPO Box 378 Canberra, ACT 2601, Australia. [5] Bocking Associates, 8 Tahlee Close, Castle Hill, NSW 2154, Australia. [6] Isotope Geology Laboratory, Boise State University, 1910 University Drive, Boise, ID 83725-1535, USA. Correspondence and requests for materials should be addressed to C.R.F. (email: cfielding2@unl.edu)

Studies of past climate transitions facilitate evaluation of future conditions in a time of rapid environmental change. The Late Permian to Early Triassic was marked by major perturbations to the Earth system, including the largest mass extinction of the geological record[1,2]. This interval saw the final assembly of Pangea, a minimum in the extent of continental flooding by the sea, the development of megamonsoonal climates, and unprecedented expansion of subtropical arid belts[1]. Geochemical proxy records reveal an Earth system susceptible to perturbation by abrupt events that may have included outgassing from the Siberian Trap Large Igneous Province (LIP)[3–5] and fly ash emission from intrusions into the West Siberian Coal Basin[6,7]. Ocean acidification[8], anoxia and euxinia[9], and ozone depletion[10] culminated in the end-Permian extinction (EPE). Such processes find parallels in the past two centuries affected by widespread deforestation, land-use changes, decline in biodiversity, and anthropogenic-induced climate change and ocean acidification, hence the signature of physical change and biotic turnover at the EPE may provide lessons for the management of modern terrestrial ecosystems.

Most research into the EPE has targeted marine records because these are perceived as the most stratigraphically complete and likely to preserve globally integrated signals[11]. However, the continental record, particularly the sedimentological and palaeovegetation signal, is receiving increasing attention because this system is predicted to be most strongly affected by future climate change and is most relevant to human society. Major biotic changes in both the marine and non-marine systems began well before the end-Permian, at the Guadalupian–Lopingian (middle–late Permian) boundary[12,13], although the severity of this early event is debated[14]. The main extinction phase is considered to have occurred shortly before the Permian–Triassic boundary and to have been short-lived (60 kyrs)[2]. However, some marine records indicate delayed extinction in deep palaeowater depths[15] and other work suggests that a second extinction episode in the earliest Triassic followed the main kill event[16,17].

The continental record of the EPE has been interpreted to signal a massive spike in soil erosion associated with devegetation, climate warming, and periods of acidic rainfall[18]. Concurrent aridification has been inferred in some cases (e.g. Karoo Basin[19,20]), despite data from mudrocks and palaeosols from that basin suggesting that earliest Triassic environments were in fact wetter than their latest Permian counterparts[21]. A long-term, global aridification trend has been inferred[18].

Several researchers have attributed an inferred change in fluvial style across the boundary from the Karoo[22,23] and other basins[24,25] to catastrophic environmental change, whereas other areas show no such change[26]. As in the marine realm, it seems that terrestrial responses to the biotic crisis were varied and complex. For example, $\delta^{13}C_{org}$, sedimentological, and palynological results from four terrestrial to marine Permian–Triassic successions of South China show that the system boundary is marked by a rapid carbon isotope shift[27], a gradual transition from fluvio-lacustrine to lagoonal-littoral deposits, strong vegetation turnover[28], and the occurrence of fungal spores together with the putative algal/fungal 'disaster taxon' *Reduviasporonites*[29], collectively interpreted to reflect a collapse of complex ecosystems and proliferation of decomposers during the EPE. Thus multidisciplinary approaches are key to interpreting environments through the Permian–Triassic transition.

Geochronological constraints on Permian–Triassic events in most continental basins studied to date are poor. In contrast, the Sydney–Gunnedah–Bowen Basin system in eastern Australia preserves a stratigraphically complete, less thoroughly documented, but well-calibrated Lopingian to Lower Triassic record that is investigated herein. The Sydney Basin of central New

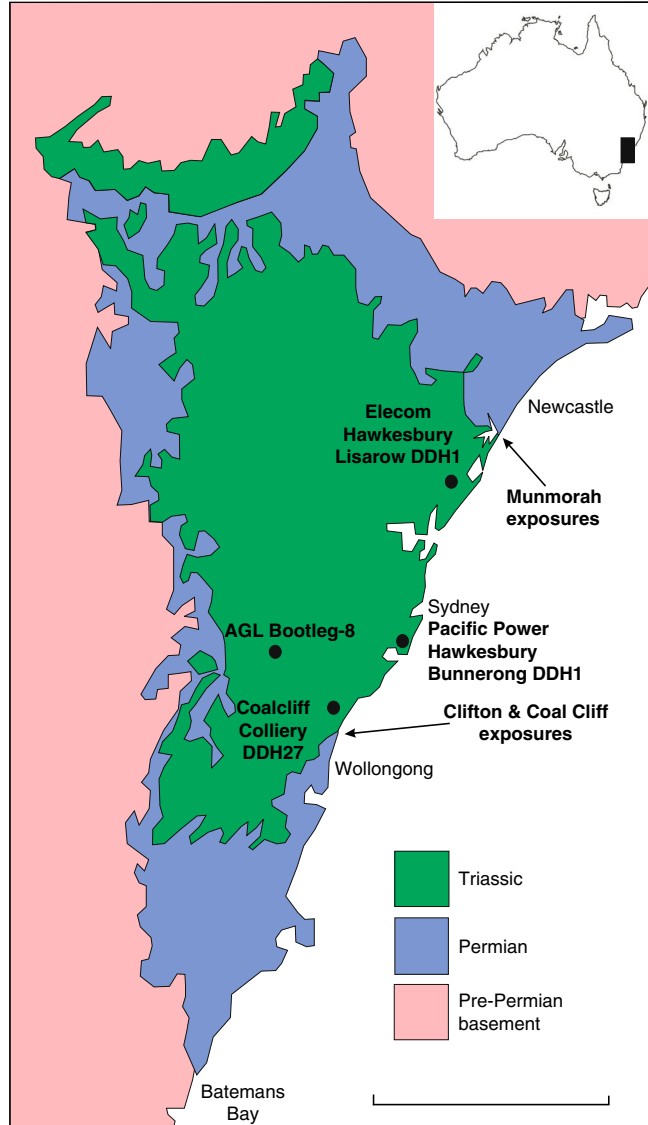

**Fig. 1** Maps of field and drillhole locations. Location of the Sydney Basin in southeast Australia and generalized geology of the Sydney Basin. Locations of field exposures and drillcores examined in this paper are also shown. Scale bar 100 km. Geology simplified and redrawn after New South Wales Geological Survey 1:500,000 Special Sheet Map of Sydney Basin (1969)

South Wales (Fig. 1) preserves a thick Permo-Triassic sedimentary succession exposed in coastal cliffs north and south of Sydney[30] and penetrated by hundreds of boreholes. Historically, the Permo-Triassic boundary (PTB) has been placed at the top of the uppermost coal seam in the succession (Bulli Coal in the south, Katoomba Coal Member in the west, and Vales Point coal seam in the north of the basin[31]). Recent high-precision chemical abrasion isotope dilution thermal ionization mass spectrometry (CA-ID-TIMS) dating of tuffs and recalibration of palynological zones in eastern Australian basins have resulted in divergent opinions about placement of the PTB with respect to local biozonations and lithostratigraphy. The PTB was equated[32,33] with the first common occurrence of *Lunatisporites pellucidus* within the Wombarra Shale or Scarborough Sandstone of the Narrabeen Group (around 40–80 m above the uppermost coal) or alternatively[34] at the top of the Illawarra Coal Measures. Regardless of the placement of the system boundary, there is a consensus that

the major crisis in terrestrial ecosystems occurred around the end of coal accumulation in the eastern Australian basins[24,33,34].

Here we use additional radiometric, palaeobotanical, and sedimentological data to identify the stratigraphic position of the end-Permian biotic crisis in a continental southern high-latitude setting and constrain its age to ~410 kyrs before age of the the PTB defined at Meishan, China. Moreover, we use the sedimentological and geochemical signals from a fully cored Permian–Triassic transitional interval in tandem with palaeoclimatic modelling to infer that the biotic crisis was accompanied by a short-term intensification of weathering within a longer-term regime of warming and amplified seasonality that played out on an alluvial landscape that experienced minimal changes in regional depositional style.

## Results and Discussion

**The Permian-Triassic boundary in the Sydney Basin**. We document a reference section for the long-term patterns of sedimentological, isotopic, floristic, and environmental change through the Lopingian and Lower Triassic succession based on the fully cored borehole Pacific Power Hawkesbury Bunnerong DDH1 (henceforth, PHKB1), drilled in the synclinal axis of the Sydney Basin near the Port of Botany, Sydney (Fig. 1). Age calibration of this southern high-latitude succession is provided by CA-ID-TIMS dating of tuffs from the present and previous studies[33,35] or, where ash beds are unavailable, by palynostratigraphic correlation to dated successions elsewhere across Gondwana (Fig. 2). The metadata for all ages are given in Supplementary document.

The boundary between the *Dulhuntyispora parvithola* and *Playfordiaspora crenulata* palynozones identified herein at the top of the Bulli Coal (and equivalents) in the Sydney Basin marks the most pronounced floristic turnover (characterized by the collapse of glossopterid mire communities) in the succession (Fig. 2) and is equated with the continental EPE herein. High-precision U-Pb CA-TIMS dating of a tuff from the lower part of the Bulli Coal in Metropolitan Colliery at Helensburgh, north of Coalcliff, has yielded an age of $252.60 \pm 0.04$ Ma (mid-Changhsingian)[33]. A new date of $252.31 \pm 0.07$ Ma (late Changhsingian) was obtained from a dark grey, organic-rich shale bed immediately overlying the Bulli Coal at Coalcliff (Fig. 2, Supplementary Fig. 1; Supplementary Tables 1 and 2). No evidence of physical working of sediment (grain-size differentiated lamination, physical sedimentary structures) was evident from this bed, and the zircon grains analysed were all euhedral and unabraded (Supplementary Fig. 2). Furthermore, volcanic fallout debris is abundant within this interval. On this basis, we argue that the age is based on primary, volcanic zircon grains and not grains reworked from older sediments.

The $252.31 \pm 0.07$ Ma age provides a constraint for the collapse of the *Glossopteris* flora, dated some 410,000 years older than the age obtained for the Global Stratotype Section and Point (GSSP) for the PTB at Meishan, China[2]. Furthermore, we have preliminary, as yet unconfirmed, ages in the range 251–252 Ma from samples within the Coal Cliff Sandstone (Fig. 2), which suggest that this unit may be of earliest Triassic age. Other authors have favoured placement of the PTB at the base, or within the lower part, of the *L. pellucidus* Palynozone[33,34,36,37], in which case the PTB would be placed around 45 m above the Bulli Coal in PHKB1, within the upper Wombarra Shale (Fig. 2).

## Sedimentology

PHKB1 terminated in diversely bioturbated, invertebrate fossil- and glendonite-bearing sandstones (Erins Vale Formation, Cumberland Subgroup, Illawarra Coal Measures) of latest middle Permian age at 1251.05 m. These are interpreted as

(glacio)marine strata of a shallow marine shelf. The uppermost indications of cold sea-floor conditions in this core are at 1212.60 m, representing the termination of the ultimate (P4) glacial interval of the Late Palaeozoic Ice Age in eastern Australia[38]. The top of the shelfal unit at 1172.56 m marks the Guadalupian–Lopingian boundary and corresponds to a major unconformity surface that can be traced across the basin[39]. The overlying Lopingian part of the Illawarra Coal Measures (Sydney Subgroup) is heterolithic and includes several coarsening-upward cycles interpreted as the record of deltaic progradational episodes. Marine invertebrate fossils are very rare in this succession, but trace fossil assemblages of reduced ichnodiversity (compared to the underlying Erins Vale Formation) indicate an array of highly stressed paralic environments[40]. Erosionally based, cross-bedded sandstone bodies of coastal plain fluvial channel origin become successively more abundant upward from about the middle of the Illawarra Coal Measures, interbedded with coal seams interpreted as the record of coastal plain mires (Fig. 2). These strata are typified by rich assemblages of *Glossopteris* leaves in finely laminated facies and by *Vertebraria* (glossopterid) roots in immature palaeosols. Channel facies in the Lopingian succession show evidence of strong seasonality in discharge, with upright in situ tree stumps in some channel-floor deposits. A restricted suite of simple, facies-crossing trace fossils characterizes these channel bodies and the interbedded heterolithic facies, and there are indicators of tidal modulation of fluvial outflow (paired mud drapes, rhythmic pinstripe lamination) at several stratigraphic levels.

The core is somewhat disturbed immediately below the level of the putative EPE; the uppermost coal (Bulli Coal) is intruded by thin dolerite sills and much of it was removed for analysis at the time of drilling. In PHKB1, and in various other sections around the Sydney Basin, the uppermost coal is succeeded by a package of 1–5 m of dark grey mudstones grading into lighter grey siltstones, which are in turn overlain by coarse-grained sandstones or conglomerates (Fig. 3). The mudstone–siltstone package includes high abundances of amorphous organic matter (AOM) and sporadic phytoplankton typical of aquatic conditions. In some parts of the basin, the sandstone/conglomerate packages lie directly on the uppermost coal with a locally scoured, but not unconformable, contact. The strata overlying the Bulli Coal indicate a continuation of similar sedimentary facies associations and depositional environments (albeit lacking coal). These overlying lithostratigraphic units (successively, the Coal Cliff Sandstone, Wombarra Shale, Scarborough Sandstone, Stanwell Park Claystone, and Bulgo Sandstone) are variably dominated by sandstones or mudrocks. Sandstone bodies are essentially unchanged in grain size, composition, internal sedimentary structures, thickness, and palaeocurrent orientations, suggesting no major change in fluvial style (mobile, sand, and gravel bed rivers) across the EPE or PTB with the exception of the termination of peat accumulation. Mudrocks remain grey or olive-grey for around 400 m into the Lower Triassic succession. Trace fossils and other coastal indicators die out upward, suggesting that the depositional environment became successively more inland fluvial in aspect over time. Approximately 400 m above the EPE horizon in PHKB1, mudrocks become conspicuously reddened (Fig. 2).

## Palaeofloras

Coal seams of the Illawarra and Newcastle Coal Measures throughout the Sydney Basin are consistently underlain by *Vertebraria* (glossopterid) roots and numerous clastic layers within and above the coals host matted *Glossopteris* leaves, indicating the extensive contribution of glossopterid gymnosperms to the (par)autochthonous accumulation of peat in the

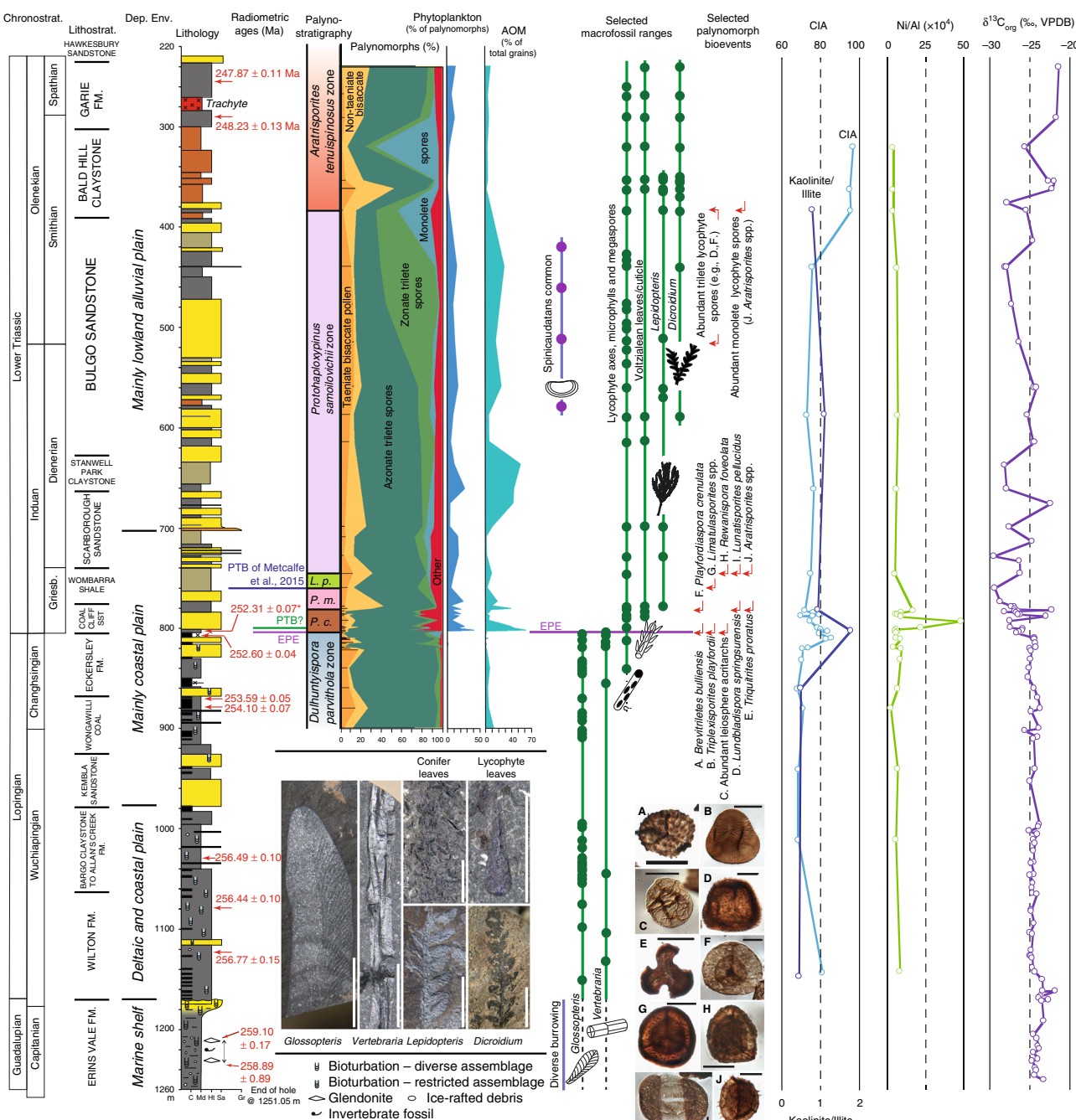

**Fig. 2** Graphical log of Pacific Power Hawkesbury Bunnerong DDH-1 (PHKB1) well core. PHKB1 is located in the synclinal axis of the Sydney Basin (Fig. 1). Stratigraphy is tied to the chronostratigraphic scale of ref. [78] (updated 2017) using new CA-ID-TIMS U-Pb ages (indicated by an asterisk) and those of refs. [33,35]. Correlation of lithostratigraphic units to all stage and substage boundaries remains tentative. Log shows lithologies (C coal, Md mudrocks, Ht heterolithic, interlaminated siltstone and sandstone, Sa sandstone, Gr conglomerate), with approximate representation of colour for mudrocks, and generalized interpretation of depositional environment. Selected plant group ranges based on macrofossils and dispersed cuticle recovered from core samples indicate the major turnover in the floras inferred to represent the EPE. First appearance datums of selected palynomorph taxa provide the basis for recognizing the local palynozones[36,37,76]. Two alternative positions are indicated for the PTB—a lower position, near the base of the *P. crenulatus* Palynozone, inferred on the basis of preliminary new CA-IDTIMS ages; and a higher position, near the base of the *L. pellucidus* Palynozone, based on ref. [33]. Scale bars for plant macrofossils = 10 mm; for palynomorphs = 20 μm. Also shown are kaolinite/illite ratios, Chemical Index of Alteration (CIA), Nickel concentration normalized to Al, and δ13Corg values through the upper Permian and Lower Triassic succession of PHKB-1. Dep. Env. = inferred depositional environment, *P. c.* = *Playfordiaspora crenulata* Palynozone, *P. m.* = *Protohaploxypinus microcorpus* Palynozone, *L. p.* = *Lunatisporites pellucidus* Palynozone, AOM = amorphous organic matter

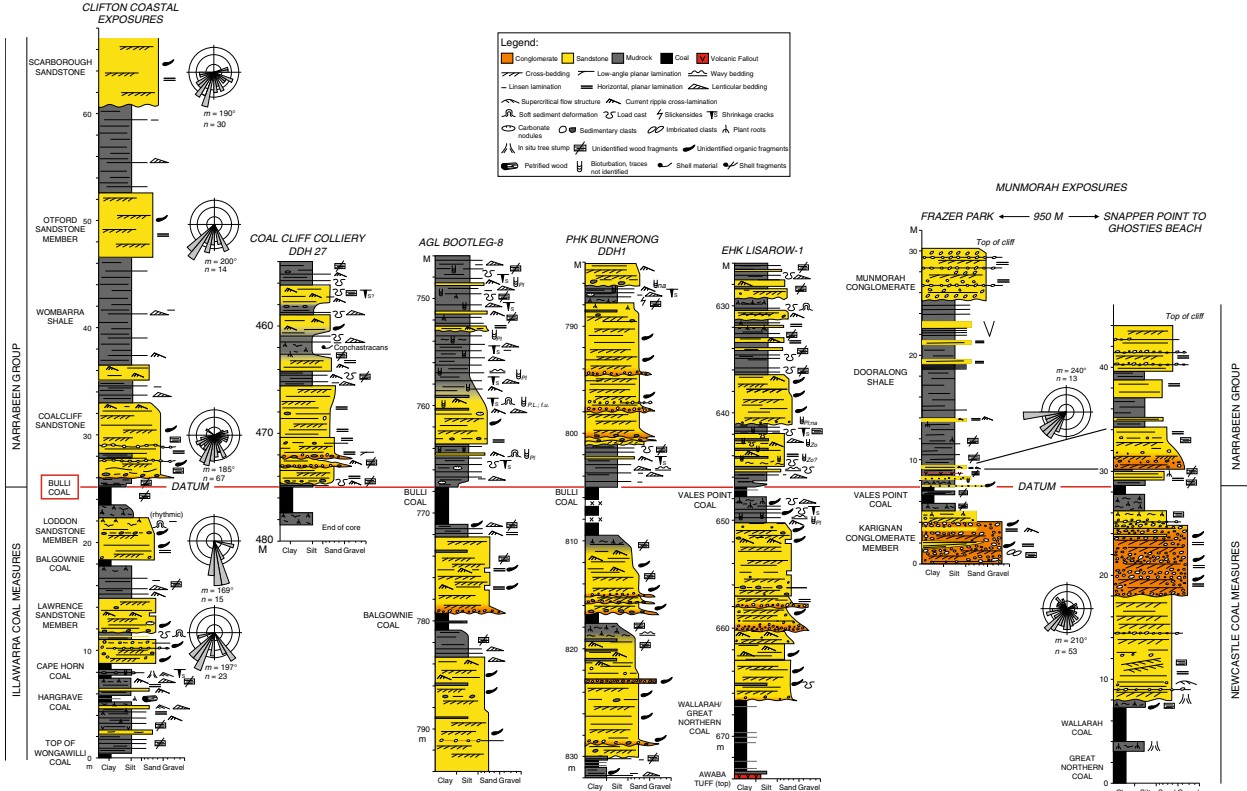

**Fig. 3** Graphic logs of the Permo-Triassic boundary interval. Several outcrop sections and drillcores across the Sydney Basin (Fig. 1) are shown, illustrating continuity in sediment body character and stacking patterns (grain size, sandstone body thickness, sedimentary structures, facies relationships) and in sediment dispersal directions, across the putative EPE horizon (red line, which is the horizontal datum for the section). Key to symbols used is also shown

latest Permian. The uppermost coal seams across the basin are all associated with a typical *Vertebraria*- and *Glossopteris*-dominated fossil flora and represent the last evidence of coal-forming environments in eastern Australia for around 5–10 million years before the re-establishment of peat-forming ecosystems in the Middle Triassic[41]. The coal measures host a diverse palynoflora of which the pollen component is dominated by *Protohaploxypinus* and *Striatopodocarpidites*. Such taeniate (striate) bisaccate pollen were typically produced by glossopterids, although a few grains of these morphotypes may derive from other gymnosperm groups. Understorey components are dominated by the spores of ferns, sphenophytes, and lycophytes. Palynofacies data are available in Supplementary Table 3. This interval is assigned to the Wuchiapingian–Changhsingian *D. parvithola* Palynozone.

*Glossopteris* leaves were not detected above the uppermost coal seam in PHKB1 but possible *Vertebraria* roots were recorded within the mudstone at 4.24 and 4.39 m above this surface. There have been assertions that elements of the *Glossopteris* flora occur at an equivalent stratigraphic level elsewhere in the basin[42], suggesting brief persistence of holdover glossopterid populations after the main 'extirpation event' (into the basal *P. crenulata* Palynozone). The 1–5-m-thick organic-rich shales overlying the Bulli Coal in the southern Sydney Basin have also yielded plant fossil assemblages marked by the first appearance of *Lepidopteris callipteroides* (Peltaspermales) and putative ginkgoaleans, voltzialean conifers, osmundaceous ferns, and isoetalean lycophytes[43] within strata we assign to the *P. crenulata* Palynozone. Both taeniate and non-taeniate bisaccate pollen in this palynozone occur in low abundance (each ~4% of the total palynomorph counts). This interval reflects rapid replacement of deciduous broad-leafed glossopterid-rich vegetation by communities

dominated by pteridophytes and sparse sclerophyllous small-leafed gymnosperms.

Successive first appearances of key pollen taxa in the 50 m of consistently fluvial facies overlying the uppermost coal in PHKB1 (*P. crenulata*, *P. microcorpus*, *L. pellucidus*, and *P. samoilovichii* palynozones: Fig. 2; Supplementary Table 4) indicate rapid turnover in the vegetation through what we interpret to be the latest Changhsingian to Griesbachian (lower Induan) stages based on far-field palynological correlation with marine sections[36].

First appearances of *L. callipteroides* and *Voltziopsis* sp. dispersed cuticle and macrofossils were recorded successively in PHKB1 at 783.2 and 779.6 m near the base of the Wombarra Shale corresponding to the upper *P. crenulata* Palynozone and the basal interval of the *Protohaploxypinus microcorpus* Palynozone, respectively. The latter zone has a palynofloral composition that is more or less the same as the preceding palynozone but with a significantly greater amount of non-taeniate pollen (*Alisporites*, *Pteruchipollenites*) as a proportion of total bisaccate pollen (~55% non-taeniate below vs ~69% above).

Above its first appearance at 745.6 m, *Lunatisporites* represents the prevalent taeniate pollen type, reflecting the persistence of floras rich in voltzialean conifers (e.g. *Voltziopsis africana*). Other well-preserved taeniate bisaccate pollen taxa occur in small numbers (average 4% of total palynomorphs) through the first three palynozones above the EPE, but the absence of glossopterid macrofossils suggests that such grains were either reworked or produced by other gymnosperm groups. Pre-EPE palynomorphs typically show evidence of extensive authigenic sulphide corrosion in the Sydney Basin, but corroded grains above this datum are scarce (Supplementary Fig. 3), hence we interpret most post-EPE taeniate bisaccate pollen to be non-reworked and of non-

glossopterid (primarily conifer or peltasperm) affinity. The increase of non-taeniate forms, mainly *Alisporites* and *Pteruchipollenites*, reflects a corystosperm or coniferous component of the vegetation that became increasingly prevalent later in the Early Triassic.

The 778–587 m interval (Wombarra Shale to lower Bulgo Sandstone) is notably depauperate in plant macrofossils apart from a few dispersed cuticle fragments of *Lepidopteris* (peltasperm seed ferns), voltzialean conifers, and lycophyte leaves. Pleuromeian lycophyte megaspores and leaves occur sporadically through the entire Lower Triassic succession but become very abundant around 614 and 537 m, respectively, in PHKB1 (both levels within the Bulgo Sandstone: *Protohaploxypinus samoilovichii* Palynozone)—an interval also marked by an increase in spinicaudatan carapaces (Fig. 2). The increase in abundance of pleuromeian megaspores (to levels >50 per 60 g sample) is matched by an increase in zonate trilete microspores (primarily *Densoisporites*) at 614 m. The latter reach a maximum abundance of 54% of the total palynomorph count at 440 m (the uppermost sample of the *P. samoilovichii* Palynozone: Fig. 2). Throughout the succeeding *Aratrisporites tenuispinosus* Palynozone, trilete zonate spores decline, and monolete zonate spores (*Aratrisporites*, which also have probable pleuromeian affinities[44]) become the dominant spores, concurrent with the deposition of the Bald Hill Claystone above the Bulgo Sandstone. *Dicroidium* (corystosperm seed fern) leaves and dispersed cuticles become common only above 370 m within the Bald Hill Claystone near the base of the *A. tenuispinosus* Palynozone. *Dicroidium* persists throughout the remainder of the Triassic in eastern Australia, becoming a dominant component of coal-forming mire ecosystems in the Middle and Late Triassic[45]. Charcoal is present in minor quantities throughout the Lopingian to Lower Triassic successions with little change in abundance.

**Geochemical proxies.** Geochemical proxy data (Supplementary Table 5) are integrated with sedimentological and floral data sets to assess changes in climate and environment through the latest Permian and earliest Triassic. The Chemical Index of Alteration (CIA), which reflects the extent of alteration of feldspars to clays, denotes changes in conditions of weathering, with higher values indicating a higher intensity of chemical weathering. All data are derived from shales. The calculation is as follows: $CIA = [Al_2O_3/(Al_2O_3 + Na_2O + K_2O + CaO^*)] \times 100$, where all oxides are in molar units and $CaO^*$ represents the CaO in the silicate fraction of the rocks[46,47]. CIA values are relatively invariant, averaging 70 through the upper Wuchiapingian and most of the Changhsingian (Fig. 2). The interval 820–720 m encompasses a two-peaked excursion to values as high as 85.7, with the highest values centring around the interval containing the uppermost coal and the loss of *Glossopteris* leaves in this core. CIA values then recover to a baseline of ~75 before rising to values that exceed 95 within the Bald Hill Claystone. The increases in CIA values around the level of the uppermost coal are indicative of a transient shift to warmer and more humid conditions that favoured increased chemical weathering. An increase in the intensity of chemical weathering across this interval is also indicated by an excursion in the ratio of kaolinite clays (dickite+halloysite+kaolinite) to illite (Fig. 2).

Throughout most of the succession, ratios of $Ni/Al$ ($\times 10^4$) values in mudrocks average 8.73 (Fig. 2). An excursion to 47.8 occurs within the Coal Cliff Sandstone, following the loss of the *Glossopteris* flora and the first appearances of several key pollen taxa (Fig. 2). Enhanced Ni concentrations have been noted across the same interval elsewhere in the southern Sydney Basin[48]. Trace element (TE)–total organic carbon (TOC) relationships indicate that the mudrocks analysed here formed mainly under dysoxic

conditions, with redox-sensitive elements most likely residing in the detrital phase (see Supplementary Fig. 4). The boundary excursion in Ni/Al occurs in mudrocks characterized by TOC < 0.15 wt%, implying contributions from an outside source rather than reflecting an increase related to development of anoxic or euxinic conditions. A similar excursion in marine sections in South China was interpreted[49] to record changes in ocean chemistry induced by Siberian Trap volcanism during emplacement of large economic concentrations of Ni in the Noril'sk region of Siberia just before and during the mass extinction interval[50]. The introduction of Ni into the marine environment was linked[49] to a rapid expansion of the methanogenic archaeon *Methanosarcina* and the delivery of a large pulse of methane to the ocean. Although Ni is an essential micronutrient, it is toxic to plants at high concentrations, negatively impacting photosynthesis and respiration, inhibiting plant growth, and causing severe depletion in plant diversity[51,52]. Enhanced Ni concentrations in the Southern Sydney Basin around the time of plant extinction and replacement implicates Siberian Trap volcanism and the release of Ni as a potential contributing factor to the end Permian die-off of terrestrial vegetation.

The carbon isotope composition of bulk organic matter ($\delta^{13}C_{org}$) is relatively invariant, averaging −24.1‰, through marine shelf and deltaic and coastal plain deposits in the lower part of the section (Fig. 2). Values become more variable in the succeeding part of the succession, with several negative excursions apparent between 610 and 801 m. The interval containing the uppermost coal and overlying siltstones is marked by a negative excursion to values as low as −27.5‰, which mirrors the spike in Ni. The overlying Wombarra Shale records an excursion to $\delta^{13}C_{org}$ values of −29.5‰. Values continue to oscillate between end-member values of −22 and −29.6‰ through the Scarborough Sandstone and Stanwell Park Claystone (Fig. 2). The Bulgo Sandstone records a long-term decrease from −25 to −28‰. The Bald Hill Claystone and Garie Formation are marked by an increase towards values of −20‰. The increased variability and negative excursions in the $\delta^{13}C_{org}$ record encompassing the loss of the *Glossopteris* flora and the subsequent turnover events imply additions of $^{13}$C-depleted carbon to the atmosphere and warming that have been documented previously in $\delta^{13}C_{org}$ records from terrestrial sites in eastern Australia[53] and elsewhere[27].

**Climate modelling.** The seasonality of the central Sydney Basin in southeastern Gondwana (at 65°–70° south palaeolatitude[54]) at the end of the Permian, as revealed from Community Climate System Model (CCSM3) simulations, was characterized by warm wet summers and cold dry winters. The maximum modelled summer-to-winter temperature difference is about 20 °C for the 4× the pre-industrial atmospheric $CO_2$ concentration and 24 °C for the 12.7× pre-industrial $CO_2$ concentration (Fig. 4a, b), potentially representing the transition from pre-EPE to PTB conditions[55]. Under these simulations, proximity to the ocean led to high precipitation during the Gondwanan summer but markedly drier conditions during the Gondwanan winter. When considering a lower cloud optical depth parameterization in the 12.7× $CO_2$ simulation, owing to biophysical-climate feedbacks in a higher $CO_2$ world[56], average temperature increases but the seasonal temperature difference decreases to about 22 °C (Fig. 4c). Significantly, under these parameters, mean winter temperatures are 9–10 °C above those for the standard 4× and 12.7× $CO_2$ simulations. This simulation might represent the Early Triassic hothouse climate[55] conducive to enhanced weathering under which small-leafed sclerophyllous evergreen trees (e.g. voltzialean conifers) may have been selectively favoured over broad-leafed deciduous gymnosperms with thin cuticle (e.g. glossopterids)

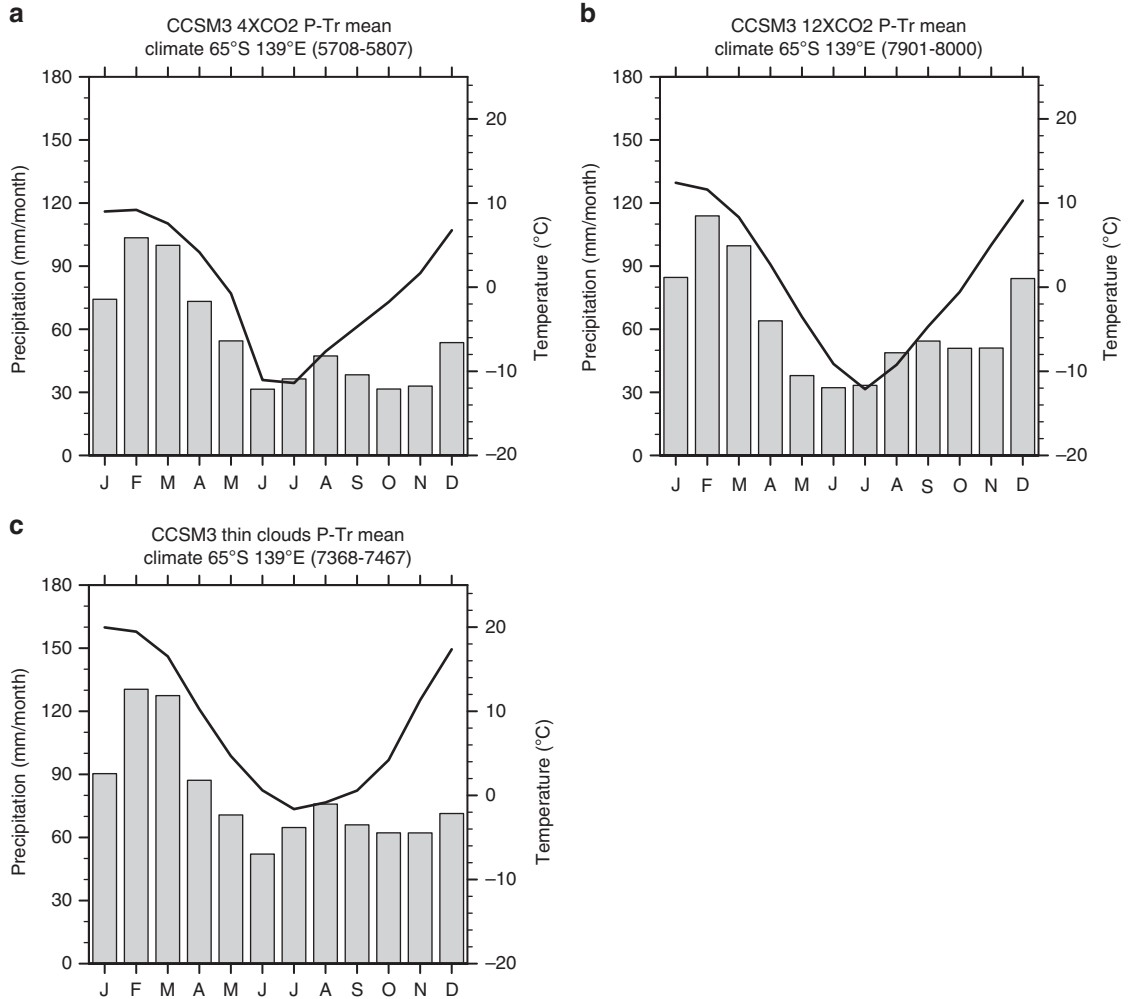

**Fig. 4** Climatographs at the palaeo-location of the Sydney Basin. Based on simulations using the CCSM3 for the PTB[55]. **a** Simulation with 4× the preindustrial $CO_2$, level, **b** simulation with 12.7× the pre-industrial $CO_2$ level, and **c** simulation with 12.7× the pre-industrial $CO_2$ level and a lower cloud optical depth

owing to more effective regulation of respiration and transpiration.

**Palaeoenvironmental interpretation.** In order to reconstruct the palaeo-landscape across the PTB, we acquired data on sandstone body thickness, grain size and composition, architecture, and palaeocurrent directions below and above the boundary from continuous surface exposures on both the southern (Illawarra Coast) and northern (Central Coast) limbs of the basin (Fig. 3). When comparing across several fluvial sediment bodies in the uppermost Illawarra Coal Measures and lowermost Narrabeen Group across the basin, we noted no upward increase in grain size or change in grain composition, no overall increase in sandstone body thickness, no perceptible change in sandstone body architecture (fluvial style), and no change in sediment dispersal directions (consistently southward) across the inferred continental EPE or PTB in any of the examined sections (Fig. 3). Furthermore, the facies immediately above the inferred continental EPE herein, both in outcrops and drillcores, are dominantly plant-fossil-bearing, grey mudrocks with mainly gradational boundaries (Fig. 3), and, in some cases, the lowermost erosionally based channel sandstone body is several metres above this level. Some localized accumulations of clay pellets and organic debris were noted above the uppermost coal[57], but these

rarely exceed a few cm in thickness and, moreover, are not dissimilar to clay-pellet granulestones lower in the Illawarra/Newcastle Coal Measures. All these features suggest that there was no exceptional erosional event or catastrophic physical degradation of the landscape at the level of the continental EPE. The predominance of grey mudrocks immediately above the Bulli Coal and equivalents across the basin, together with sporadic phytoplankton and AOM within these strata, suggests that the extinction interval was characterized by generally moist conditions even after the loss of peat-forming floras. Furthermore, the change in mudrock colour from dominantly grey to brown is gradual over 10s to 100s of metres of section above the level of the continental EPE, suggesting that a change to more freely drained alluvial landscapes was a progressive, long-term transition, rather than an abrupt change.

The simulated gradual climate changes in this region, even taking into consideration major injections of $CO_2$ into the atmosphere, are consistent with the finding that sedimentation patterns did not change significantly over the investigated period. The simulated warmer and wetter conditions in the Sydney Basin area for the scenario potentially representing Early Triassic conditions can be explained by its proximity to the ocean. These results are in agreement with the higher CIA values for that time, indicating increased weathering relative to the late Permian.

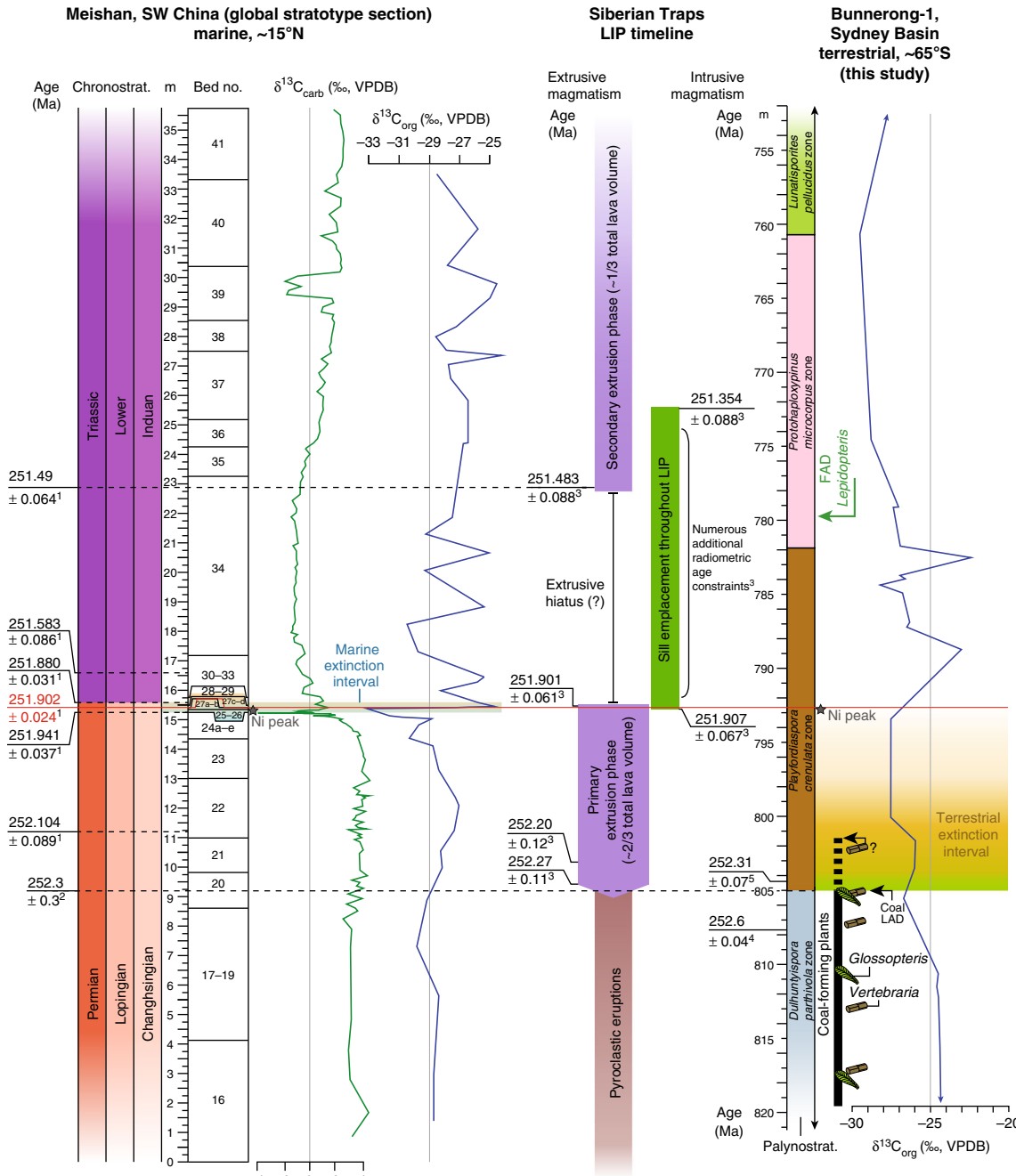

**Fig. 5** Age correlations of the PTB and EPE. Correlations shown between the marine Permian–Triassic global stratotype section at Meishan, the terrestrial Sydney Basin succession, and the magmatic phases of the Siberian Traps Large Igneous Province. All absolute age uncertainties are at 2σ level; 1 = ref. [2], 2 = ref. [79], 3 = ref. [50], 4 = ref. [33], 5 = this study. FAD = first appearance datum, LAD = last appearance datum. Palaeolatitudes from ref. [54]

In contrast to the consistency of sedimentary style and moderate climatic changes in this area, the plant macrofossil and palynological records indicate major shifts in plant group representation through this interval. Most notable is the abrupt collapse of glossopterid mire ecosystems at the inferred level of the continental EPE, with persistence of just a few survivors (e.g. a single *Glossopteris* species and *Schizoneura gondwanensis*) elsewhere in the basin for only a few metres above the boundary. Our data indicate that the primary palaeobotanical extinction event took place between 700 and 410 kyrs prior to the PTB, consistent with recent findings from China, and ~370 kyrs before the main marine extinction recorded in Meishan Bed 25 now dated at 251.941 ± 0.037 Ma[2,58,59] (Fig. 5). This collapse was succeeded in

the PHKB1 drillcore by a succession of relatively short-ranging plant communities (represented by the *P. crenulata*, *P. microcorpus*, *L. pellucidus*, and *P. samoilovichii* palynozones) variably dominated by peltasperm seed ferns, voltzialean conifers, and pleuromeian lycophytes through the latest Permian and earliest Triassic. Communities dominated by corystosperm (Umkomasiales) seed ferns did not become established until late in the Early Triassic (around 3 m.y. after the EPE[60]: Fig. 2). The vegetation signal suggests the influence of far-field processes that overprinted the relatively stable regional (basin-wide) landscape conditions. The CIA, kaolinite/illite, and $\delta^{13}C_{org}$ records indicate that the continental EPE was marked by a brief perturbation to warmer, more humid climate conditions that led to an increase in

chemical weathering. Despite the high palaeolatitude (65–70°S[54]), there is no evidence for a short-lived PTB ice age as proposed recently[61].

The last evidence of cold-water indices (P4 interval of the Late Palaeozoic Ice Age) around the Guadalupian–Lopingian boundary, disappearance of coal-forming mires near the close of the Permian, and intensification of redbed development together with very high CIA values in the late Early Triassic appear to reflect a very long-term trend towards warming over the 10 m.y. from the Wuchiapingian to mid-Olenekian. Moreover, this general pattern is reflected across the higher-latitude regions of the Southern Hemisphere, where glossopterid-dominated communities are consistently replaced around the close of the Permian by successive peltasperm-, voltzialean conifer-, pleuromeian lyco-phyte-, and corystosperm-dominated vegetation, with widespread representation of redbed successions 10s to 100s of metres above the level of the continental EPE[62,63]. Further, there may have been a rapid poleward progression in the replacement of these vegetation types with glossopterid coal-forming communities disappearing significantly earlier in northern Gondwana compared to southern palaeolatitudes[64]. The youngest examples of glossopterid leaves occur in strata of East Antarctica very close to the PTB[65,66], suggesting that south polar regions acted as the final refugium for cool-climate, hygrophilous glossopterid gymnosperms[67]. Successive plant communities of the latest Permian and earliest Triassic probably initiated and collapsed upon reaching key environmental thresholds that were linked to temperature, the availability of moisture, the intensity of climatic seasonality, or levels of environmental contaminants (e.g. nickel or ozone-depleting halogens). Accurately dating the changes, quantifying these environmental tipping points, and defining the precise drivers of the global-scale environmental changes are key targets for future research, since they are relevant to understanding biotic responses to abrupt anthropogenic warming in the modern world.

**Timing of the terrestrial end-Permian extinction event.** The Permian–Triassic boundary GSSP at Meishan, China records two distinct pulses of marine faunal extinctions close to 251.941 ± 0.037 and 251.880 ± 0.031 Ma[2]. These discrete events define the boundaries of a marine 'extinction interval' (sensu ref.[17]), which corresponds to stable carbon isotope excursions[68,69] and a Ni abundance spike[70]. The age of this extinction interval is well correlated to the onset of massive magmatic intrusion of the Siberian Traps LIP[50]. In contrast, the floral collapse evident in the Sydney Basin is constrained by two U-Pb age high-precision determinations: 252.6 ± 0.04 and 252.31 ± 0.07 Ma. As such, this terrestrial extinction interval, represented by a collapse of the *Glossopteris* flora, cessation of coal-forming conditions, and enhanced levels of phytoplankton and AOM, occurred ~370 kyrs prior to the onset of the marine extinction interval. The terrestrial extinction in the Sydney Basin is approximately concurrent with the onset of the primary extrusion phase of the Siberian Traps LIP[50] (Fig. 5). Siberian Traps volcanism is also implicated by severe fluctuations in stable carbon isotopes and an increase in Ni abundance during the terrestrial extinction interval (*P. crenulata* Palynozone; Fig. 5). This suggests that, despite a temporal decoupling of the terrestrial and marine biotic collapses, Siberian Traps magmatism was the likely trigger in both realms.

## Methods

**Geochronology.** Zircon grains were separated from a dark grey mudrock (sample GA2167007) using standard techniques. The sample contains round grains that appear to be detrital zircon and sharply faceted grains that could be primary volcanic zircon. Only the sharply faceted grains were picked. Grains were annealed at 900 °C for 60 h in a muffle furnace. They were then mounted in epoxy and polished to expose their centres. A JEOL JSM-1300 scanning electron microscope coupled with a Gatan MiniCL were used to obtain cathodoluminescence photographs of grains (Supplementary Figure 2). Zircons were analysed by laser ablation inductively coupled plasma mass spectrometry (LA-ICPMS) with a ThermoElectron X-Series II quadrupole ICPMS and New Wave Research UP-213 Nd:YAG UV (213 nm) laser ablation system (Supplementary Table 1). Acquisition and calibration of U-Pb dates and a suite of high field strength elements (HFSE) and rare earth elements (REE) utilized in-house analytical protocols, standard materials, and data reduction software. A 25-μm-wide laser spot was used to ablate zircons. Fluence and pulse rates of 5 J/cm$^2$ and 10 Hz, respectively, were used for 45 s (15 s gas blank, 30 s ablation) to excavate ~25 μm deep pits. A 1.2 l/min He gas stream carried the ablated material to the nebulizer flow of the plasma. Dwell times were as follows: 5 ms for Si and Zr, 200 ms for $^{49}$Ti and $^{207}$Pb, 80 ms for $^{206}$Pb, 40 ms for $^{202}$Hg, $^{204}$Pb, $^{208}$Pb, $^{232}$Th, and $^{238}$U and 10 ms for all other HFSE and REE. Prior to each analysis, background count rates for each analyte were obtained and recorded. These were subsequently subtracted from raw count rates for each analyte. Pits that intersected glass or mineral inclusions within zircons were identified based on concentrations of Ti and P. U-Pb dates from these analyses are considered valid in cases where it is evident that U/Pb ratios were not affected by inclusions. Those analyses that show evidence of contamination by common Pb (mass 204 being above a baseline) were rejected. To calculate concentrations of each analyte, background-subtracted count rates were normalized internally to $^{29}$Si and calibrated relative to primary standards NIST SRM-610 and -612 glasses. Temperature calculations are based on the Ti-in-zircon thermometer (Supplementary Table 1). An average value in crustal rocks of 0.8 was used because of the lack of constraints on the activity of TiO$_2$.

In determining U-Pb and $^{207}$Pb/$^{206}$Pb dates, the effects of instrumental fractionation of the background-subtracted ratios was corrected and dates were calibrated relative to regular measurements of zircon standards and reference materials. Time-dependent instrumental fractionation was monitored by including two analyses of the primary standard, the Plešovice zircon[71], for every ten analyses of an unknown zircon. Results of measurements from the zircon standards Seiland (530 Ma, data available from Boise State University) and Zirconia (327 Ma, data available from Boise State University), treated as unknowns and measured once for every 10 analyses of an unknown zircon, were used to generate secondary corrections to $^{206}$Pb/$^{238}$U dates. A linear age bias of several percent, related to the $^{206}$Pb count rate, was evident. The secondary correction was used to mitigate matrix-dependent variations due to contrasting compositions and ablation characteristics between the Plešovice zircon and other standards (and unknowns).

All analyses include uncertainties related to radiogenic isotope ratio and age error propagation from counting statistics and background subtraction. First, Isoplot 3.0 (http://www.bgc.org/isoplot_etc/isoplot.html) was used to calculate a weighted mean $^{206}$Pb/$^{238}$U date from equivalent dates (probability of fit >0.05), using errors on individual ages that do not include uncertainties related to standard calibration. Subsequently, a standard calibration uncertainty is propagated into the error on the weighted mean age. This uncertainty (1.2%; 2σ) equates to the local standard deviation of the polynomial fit to the interspersed primary standard measurements vs time. Errors on the dates are 2σ.

The CA-TIMS method was used to obtain U-Pb ages from analyses on single zircon grains (Supplementary Table 2). Zircons for U-Pb age analysis were selected based on U-Pb dates from LA-ICPMS.

In preparation for analysis, zircon grains were placed in 3 ml Teflon perfluoroalkoxy (PFA) beakers, loaded into 300 μl Teflon PFA microcapsules, and placed in a large-capacity Parr vessel capable of holding 15 microcapsules. Zircon samples were placed in 120 μl of 29 M hydrofluoric acid (HF) for 12 h at 190 °C and partially dissolved. Etched zircon was then returned to 3 ml Teflon PFA beakers from which HF was decanted. Zircons were subsequently immersed in 3.5 M HNO$_3$, placed in an ultrasonic bath for 1 h, and then fluxed for an hour at 80 °C on a hotplate. Following this procedure, HNO$_3$ was removed and zircon was rinsed twice in ultrapure H$_2$O prior to reloading into 300 μl Teflon PFA microcapsules that had been rinsed and fluxed in 6 M hydrochloric acid (HCl) during the sonication and washing process and spiked with the EARTHTIME mixed $^{233}$U–$^{235}$U–$^{205}$Pb tracer solution. Zircon was then dissolved in Parr vessels in a solution consisting of 120 μl of 29 M HF and a trace of 3.5 M HNO$_3$ for 48 h at 220 °C. Samples were dried to fluorides and re-dissolved in 6 M HCl overnight at 180 °C. An HCl-based anion-exchange chromatographic procedure[72] was used to separate U and Pb from the zircon matrix. U and Pb were then eluted together and dried with 2 μl of 0.05 N H$_3$PO$_4$.

In preparation for analysis, Pb and U were loaded on a single outgassed Re filament in 5 μl of a silica-gel/phosphoric acid mixture[73]. U and Pb isotopic measurements were determined using a GV Isoprobe-T multicollector thermal ionization mass spectrometer equipped with an ion-counting Daly detector. Measurement of Pb isotopes was carried out by peak-jumping all isotopes on the Daly detector for 160 cycles. Results were corrected for 0.16 ± 0.03%/a.m.u. (1σ error) mass fractionation. Transitory isobaric interferences brought about by the presence of high-molecular weight organics, particularly on $^{204}$Pb and $^{207}$Pb, disappeared within approximately 30 cycles as ionization efficiency averaged 104 cps/pg of each Pb isotope. Analysis of NBS982 was used to determine linearity (to ≥1.4 × 106 cps) and the accompanying deadtime correction of the Daly detector. Uranium was analysed as UO$_2{}^+$ ions in static Faraday mode on 1012-ohm resistors for 300 cycles and corrected for isobaric interference of $^{233}$U$^{18}$O$^{16}$O on

$^{235}U^{16}O^{16}O$ with an $^{18}O/^{16}O$ of 0.00206. For each U isotope, ionization efficiency averaged 20 mV/ng. The known $^{233}U/^{235}U$ ratio of the EARTHTIME tracer solution was used to correct for U mass fractionation.

Published algorithms, EARTHTIME ET535 tracer solution[74] with calibration of $^{235}U/^{205}Pb = 100.233$, $^{233}U/^{235}U = 0.99506$, and $^{205}Pb/^{204}Pb = 11268$, and U decay constants (Supplementary Table 2) were used to calculate CA-TIMS U–Pb dates and uncertainties. $DTh/U = 0.20 \pm 0.05$ ($1\sigma$) and published algorithms (Supplementary Table 2) were used to correct $^{206}Pb/^{238}U$ ratios and dates for initial $^{230}Th$ disequilibrium. This resulted in an increase of ~0.09 Ma in the $^{206}Pb/^{238}U$ dates. All common Pb in analyses was attributed to laboratory blank and subtracted based on the measured laboratory Pb isotopic composition and associated uncertainty. U blanks are estimated at $0.013 \pm 0.009$ pg ($1\sigma$).

Isoplot 3.0 was used to calculate weighted mean $^{206}Pb/^{238}U$ dates from equivalent dates (probability of fit > 0.05). Errors on the weighted mean dates are $\pm x/y/z$. In this equation, $x$ is the internal error based on analytical uncertainties only, including subtraction of tracer solution, counting statistics, and blank and initial common Pb subtraction. The $y$ factor includes the tracer calibration uncertainty propagated in quadrature, whereas $z$ includes the $^{238}U$ decay constant uncertainty propagated in quadrature. When comparing our dates with $^{206}Pb/^{238}U$ dates from other laboratories that used the same EARTHTIME tracer solution or a tracer solution that was cross-calibrated using EARTHTIME gravimetric standards, internal errors should be considered. In addition, errors such as the uncertainty in the tracer calibration should be considered when comparing our dates with those derived from other geochronological methods (e.g. LA-ICPMS) that use the U–Pb decay scheme. Errors including uncertainties in the tracer calibration and $^{238}U$ decay constant (Supplementary Table 2) should also be considered when comparing our dates with those derived from other decay schemes (e.g. $^{40}Ar/^{39}Ar$, $^{187}Re-^{187}Os$). Errors for weighted mean dates and dates from individual grains are given at $2\sigma$.

Nineteen grains from 2,167,007 analysed by LA-ICPMS yielded dates between $426 \pm 17$ and $243 \pm 10$ Ma. The youngest nine dates are equivalent with a weighted mean of $252 \pm 5$ Ma (MSWD = 1.9, probability of fit = 0.06). Seven of the grains yielding young dates were removed from the mounts and analysed by CA-TIMS. Two of the grains were broken into two fragments that were analysed separately. The nine dates (Supplementary Figure 1) are equivalent with a weighted mean of 252.31 ± 0.07/0.14/0.30 Ma (MSWD = 1.0, probability of fit = 0.48). Despite the large population of detrital zircon in this sample as indicated by >260 Ma LA-ICPMS dates and round grains, the CA-TIMS weighted mean date is interpreted as the depositional age because a relatively large number of grains yielded equivalent dates.

**Palynology.** Fifty-two samples were chosen from well core sediment samples of PKHB1 for palynological processing and slide preparation. These were digested using HCl and HF. For palynofacies, kerogen slides were produced for all samples. For palynomorph counts, 44 samples were oxidized with Schulze's Solution, then sieved at 10 μm. All slides are provided with prefix 'S' and stored at the Department of Palaeobiology, Naturhistoriska riksmuseet, Stockholm, Sweden. Palynomorph images were collected on a Zeiss Axioskop 2 Plus transmitted light microscope, and images were taken on a Zeiss AxioCam MRc camera.

Palynofacies data were compiled from counts of 500 individual grains (minimum grain diameter = 5 μm). The following palynofacies categories were included in the counts (following the classification of Tyson, 1995[75]): (1) plant spores, (2) pollen, (3) phytoplankton, (4) opaque phytoclasts, (5) tracheids/rays, (6) other translucent phytoclasts, (7) cuticles/membranous tissues, (8) particulate AOM, (9) fungal debris, and (10) resin. Palynomorph counts consisted of 250 grains, but 6 of the samples counted for palynomorphs failed to meet the total specimen count (S014097, S014099, S014100, S014105, S014116, and S014141). Where present, index spore-pollen taxa were counted from both sieved and kerogen slides. Biostratigraphic correlations were based on regional zonation schemes[36,37,76]. The details for all samples examined in this study are summarized in Supplementary Tables 3 and 4.

A qualitative taphonomic assessment of spores and pollen was included to demonstrate the degree of reworking within this section. In this case, we applied the preservation quality of pre-EPE spores and pollen fossils vs those from post-EPE strata. Pre-EPE specimens typically have a high degree of corrosion. The style of degradation is characteristic of authigenic sulphide mineralization (e.g., pyrite, marcasite). As authigenic sulphides have been shown to form within organic-rich reducing conditions during or soon after sedimentation, this diagenetic effect on pre-EPE specimens was likely (pene)contemporaneous. Thus this degradation pattern serves as a proxy for specimens that have been reworked from pre-EPE sediments into overlying strata. This effect is not likely a local phenomenon as it has been noted in other wells from the Sydney Basin (Coalcliff Colliery DDH 27, EHK Lisarow-1; Fig. 3). However, the occurrence of such degraded grains in post-EPE strata is negligible, indicating only minor reworking. We present the photographic evidence in Supplemental Fig. 3 to support this interpretation.

**Geochemistry.** Elemental concentrations in shales and mudrocks (powdered) were determined by both X-ray fluorescence (XRF) and ICP-atomic emission spectroscopy (AES). XRF analyses were carried out on pressed powders using a Bruker Tracer 5i equipped with an Rh X-ray tube at the University of Nebraska-Lincoln. Each sample was analysed at low-energy (major elements) and high-energy (TEs) conditions for count times of 15 and 60 s, respectively. Low-energy analyses were

undertaken at 15 kV with no filter, whereas high-energy analyses were undertaken at 40 kV with a Ti25um Al 300 filter. ICP-AES measurements were conducted by American Assay Laboratories on 0.5 g samples using a five-acid digestion. CIA was calculated using Al, Ca, K, Na, and P concentrations by ICP-AES and Si concentrations by XRF (Supplementary Table 5). Ni/Al ratios shown in Fig. 2 were determined via ICP-AES.

The stable carbon isotope composition of bulk organic matter ($\delta^{13}C_{org}$) was determined on splits of samples used for palynological analysis (Supplementary Table 3). TOC, total nitrogen, and $\delta^{15}N$ values were measured simultaneously but not reported here. In preparation for analysis, samples were powdered. Up to 500 mg of sample was placed in a 50 ml centrifuge tube and reacted overnight with 1 N HCl at room temperature to remove carbonate mineral phases and rinsed three times with ultra-pure water. In each case, the supernatant was separated by centrifugation (3000 RPM for 5 min) and discarded. Samples were subsequently dried and crushed using an agate mortar and pestle. Sample splits were analysed using a Costech Elemental Analyser connected to a Thermo Finnigan MAT 253 stable-isotope gas-ratio mass spectrometer at the Keck-NSF Palaeoenvironmental and Environmental Laboratory at the University of Kansas. Carbon isotope compositions are reported in per mil (‰) relative to the Vienna Peedee Belemnite (V-PDB) standard. Montana Soil (NIST Ref Mat. 2711) and a calibrated yeast standard were used to monitor quality control, with compiled results over 2 years showing analytical error to be within ±0.22‰ for $\delta^{13}C$.

**Palaeoclimate modelling.** The climate transition across the PTB was simulated by three time-slice experiments with the global Community Climate System Model version 3 (CCSM3)[77]. The atmosphere was resolved by a horizontal spectral T31 grid (~3.75° × 3.75°) and by 26 unevenly spaced terrain-following vertical layers, while the ocean had a nominal 3° horizontal grid and 25 vertical layers with higher resolution near the surface. The boundary conditions consisted of a palaeo-insolation of 1338 W/m² (which is ~2.1% lower than the present value), palaeo-orbital settings (eccentricity of 0°, obliquity of 23.5°), palaeogeography, and palaeo-land surface conditions[55].

Three climate simulations have been carried out to assess the environmental changes across the PTB. For a baseline scenario for the time prior to the flood basalt volcanism, greenhouse gas levels of four times the pre-industrial level of 280 ppmv ($4\times CO_2$) were used[55]. The second climate scenario assumed elevated atmospheric greenhouse levels of $12.7\times CO_2$ for near-PTB conditions. In addition, an Early Triassic hothouse climate sensitivity experiment at $12.7\times CO_2$ considered a reduced cloud optical thickness in response to changes in the cloud droplet radii and concentration[55,56]. All simulations were equilibrated into quasi-steady state (with ~5500–8000 years of integration).

**Code availability.** The climate model description and code are available at http://www.cesm.ucar.edu/models/ccsm3.0/

**Reporting summary.** Further information on experimental design is available in the Nature Research Reporting Summary linked to this article.

## Data availability
The authors declare that all data supporting the findings of this study are available within the paper and its supplementary information files. Palynological slides used in this study are held in the collections of the Swedish Museum of Natural History (NRM), Stockholm under the registration codes SMNH S014095–014149. Core samples and plant fossil records are held in the W.B. Clarke Geoscience Centre drillcore library, Londonderry, New South Wales, Australia, with duplicate photographic records held at NRM. A reporting summary for this article is available as a supplementary information file.

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

## Acknowledgements

This research was funded by collaborative research grants from the National Science Foundation (EAR-1636625 to C.R.F. and T.D.F. and EAR-1636629 to A.W. and C.W.). Funding was also received from the Swedish Research Council (VR grant 2015-4264 to V.V., and VR grant 2014-5234 to S.M.).

## Author contributions

C.R.F., T.D.F., S.M., V.V., C.M., R.S.N., and M.B. examined and sampled the PHKB-1 core and performed field investigations. C.R.F. contributed to sedimentology; T.D.F. and A.P.T. geochemistry; S.M., V.V., and C.M. palaeobotany and palynology; R.S.N. and J.L. C. were responsible for geochronology, and A.W. and C.W. performed climate model simulations. C.R.F. compiled the paper and all authors contributed to manuscript refinement.

## Additional information

**Competing interests:** The authors declare no competing interests.

