## [Peer Review File · Nature Communications]

Reviewers' Comments:

Reviewer #1:

Remarks to the Author:

Fascinating, highly significant and much needed re-evaluation of the high latitude Permo-Triassic terrestrial record of Australia. Conclusions are highly significant – especially the lack of climate change across the extinction interval – contrasts with earlier claims from other regions. The precision dating also reveals the extinction pre-dates the marine extinction, as already inferred for lower latitude settings, again an important finding. The possibility of a diachronous *Glossopteris* extinction from low to high latitude is also potentially important.

I just have a few minor points:-

“Such a catastrophic and abrupt environmental change, if repeated, would have devastating consequences for life on Earth.” – self-evident therefore not needed, and bordering on pretentious.

“well-calibrated Lopingian and Lower Triassic” I would replace ‘and’ with ‘to’

“Bulli seam in the south, Katoomba seam in the west, and Vales Point seam in the north of the basin” – shouldn’t seam be a capital ‘Seam’ as it is a proper noun?

Figure 2 – somewhere in the caption it needs to be stated that this a core record.

“persistence of refugial glossopterid populations after the main ‘kill- event’ (into the *P. crenulata* Palynozone).” – delete “refugial” a loaded term implying migration to a safe area which is inappropriate here, “holdover” is the correct term needed here.

As written the “Palaeofloras” section seem to record higher diversity above the extinction horizon than before it – see also figure 2 – would it be best to describe this as a turnover event rather than an extinction event?

“not dissimilar to clay-pellet granulestones” – what’s the origin of these, if not reworking?

“The predominance of gray mudrocks immediately above the Bulli Coal and equivalents across the basin, together with sporadic phytoplankton cysts and AOM within these strata,” doesn’t this suggests marine influence, otherwise where do the phytoplankton come from? Was the Bulli Coal forest drowned/ transgressed?

“by relatively short-ranging plant communities” – this needs clarifying because the next sentence then says that the next community appears 3 myr later, so the “short ranging” sounds like a very long time.

Reviewer #2:

Remarks to the Author:

Fielding and co-authors present a multi-proxy study of the Bunnerong DDH-1 core from Sydney Basin. The core is about 1 km thick, and intersects the Permian-Triassic Boundary and mass extinction in a terrestrial sedimentary setting. Work on the core includes a sedimentologic log, palynology, major (CIA) and minor (Ni) element geochemistry, organic carbon isotopes, and the development of a single

new U-Pb age via CA-ID-TIMS. I am not a paleontologist or paleobotanist, and so the most relevant aspect of my background is my experience with U-Pb geochronology. With regards to U-Pb, I see no potential issues. It comes from a top-shelf laboratory in BSU, and the data look great. The development of $\delta^{13}\text{C}_{\text{org}}$ records is also fairly standard, and cannot see any reason why the data is not reasonable.

But as a non-specialist, it is not clear to me how this work represents a major development in the study of the P-T mass extinction that would be appropriate for a generalist journal like Nature Communications. For example, the conclusions presented here appear no different from Metcalfe et al. (2015), which this MS cites and shares several co-authors with. In that contribution to Gondwana Research, 28 CA-ID-TIMS ages are presented from the Sydney and Bowen basins, as oppose to only one presented here. In their Fig. 15, Metcalfe et al. (2015) place the terrestrial extinction horizon right above the Bulli Coal, just like this MS; this figure also shows many of the same paleontologic/paleobiologic trends (loss of Glossopterids and coals, palynologic turnover) as that boundary that are discussed in this MS. Fig. 16 of Metcalfe et al. (2015) also summarizes $\delta^{13}\text{C}_{\text{org}}$ chemostratigraphies from across Australia, and they do not look substantially different from what is presented here.

The single age developed here is a nice, new constraint to have, as it sits right during the start of the mass extinction event and thereby improves the precision offered of the extinction age. The CIA and Ni records are also new, and seem concordant with predictions about Earth system conditions predicted during the P-T extinction. The data should be published, but unless the authors can articulate more clearly why their work represents a significant advance that would appeal to a general audience, I think it should be submitted to a more specialist journal.

A couple of comments:

-the climate modelling is barely discussed in this paper; the results seem entirely confined to the Results section, and do not come into the discussion really at all. I suggest removing it OR expanding it to become better integrated with the rest of the MS

-in future versions, include line numbers on the text. It is very hard to provide specific comments otherwise. I have made some directly onto the PDF itself.

Jon Husson

Reviewer #3:

Remarks to the Author:

The contribution by Fielding and colleagues focuses primarily on the Pacific Power Hawkesbury Bunnerong drill core, supplemented with a sedimentological context from correlative outcrop exposures, in which they present stratigraphic/sedimentologic, geochemic, palynologic, and geochronometric data, which are then used to model paleoatmospheric conditions in the latest Permian and earliest Triassic of eastern Australia. The authors proposed the onset of a short-lived climate perturbation approximately 370 ka prior to the onset of the end-Permian marine extinction event, wherein the demise (collapse) of the Glossopteris flora is proposed in this high paleolatitudinal position. This extirpation, rather than extinction (if other southern hemisphere records are accurate about the continuance of the flora in Antarctica), of the group in the Sydney Basin is used as a potential model for how vegetation might respond to climate-warming scenarios proposed for our current world.

What is exciting about the collective work of this research group is the multidisciplinary approach in a part of the world where much has been claimed by others. In many of the previous publications, few hard details have been placed into any high resolution lithostratigraphy constrained by geochronology, other than the work of Metcalfe and colleagues, whose interpretations about the placement of the end-Permian extinction and palynological biozones continue to be open to discussion. The fact that there is a notable loss of the glossopterid biome subsequent to the youngest peat swamp accumulation has been taken to represent a terrestrial reorganization as a consequence of some global perturbation.

The PHKB1 reference core is a little over 1000 m in length and, overall, is a prograding system from offshore marine shelf ending in "mostly" lowland alluvial plain settings. Peat (Bulli coal seam) accumulated in a coastal plain environment and palynological data from this interval shows a marked turnover in floral elements and "collapse" of the glossopterid flora. Euhedral zircons recovered from immediately above the coal, and analyzed by CA-ID-TIMS, yield a concordant date of 252.31 +/- 0.07 Ma, and these data are used to identify the continental End Permian Mass Extinction event.

This reviewer has several questions, most of which can be answered but one of which can't be addressed; that is, the "somewhat disturbed" nature of the core at the putative EPME. This latter concern will be addressed first. Nowhere in the text nor the supplemental materials is the "somewhat disturbed" nature described or presented. What, exactly, do the authors mean? This reviewer suspects that the coal (how much of its thickness, for example) had been removed from standard proximal analyses to assess its quality. And, if this horizon is identified as the point of turnover, is that based on the fact that the floristic elements in the peat are presumed to be glossopterid-dominated? Presumably, the Bulli coal is mined extensively elsewhere in the basin, and this reviewer wonders why a channel sample of that coal in a geographically close mine site wasn't sampled, processed, and used as a check on the core data. The overlying mudrock preserves marine acritarchs which indicate retrogradation in this part of the section; might a correlative site where coastal plain deposits that overlie the Bulli coal be appropriate to sample and assess, to determine whether there is such an "abrupt" loss that might not be a consequence of coastal plain inundation? The authors identify a downcutting event above the Bulli coal in other parts of the basin, with the presence of a conglomeratic sandstone in erosional contact. They also use this evidence, in combination with geochemical data, to demonstrate a continuation of the same fluvial and pedogenic style across the landscape following peat termination.

A second question arises about the palynological data, herein presented in a very generalized manner that is not wholly fulfilling nor transparent. The authors present pollen diagrams in which the proportion of spores vs. pollen are illustrated, along with aspects of phytoplankton and the dispersed organic matter (palynofacies; Fig. 2). Yet, the claim is that there is a collapse of the glossopterid flora without presenting the data to substantiate it. For example, why not provide the reader with plots of the percentage of glossopterid palynological components versus the percentage of pollen in each sample? Just plotting total spores vs. pollen tells the reader nothing about how the proportion of glossopterid representation has changed over the proposed "extinction" interval. The authors note that they do continue to find "small" amounts of glossopteris pollen in overlying sediments, and attribute these (unspecified numbers) to having been reworked. The question of how much reworking of slightly older palynomorphs into slightly younger sediments is always a difficult problem to solve, mainly because there are few, if any, physical characteristics to separate these two populations. The reader is not informed about what percentage of glossopterid pollen is considered to represent such reworked palynomorphs. And, more importantly, the authors provide no statistical test to demonstrate the high probability of these "low numbers" as having been reworked, nor do they specify the interpreted depositional environment from which the palynomorphs were recovered.

How to test for "reworking" of glossopterid pollen? This reviewer suggests the possibility of using the proportion of glossopterid pollen recovered from siliciclastic deposits below and above the Bulli coal (but not from the coal, itself). They state that taeniate and non-taeniate bisaccate pollen average 3.9% and 4.4% in the lithologies above the Bulli coal. What are their proportions in siliciclastic lithologies below the Bulli coal? If the pre-event proportions are statistically different from the post-event proportions in the same depositional setting, controlling for potential taphonomic biases, then one could argue that the same proportion of glossopterid contribution outside of the peat swamps was occurring. In contrast, if the pre-event proportions are statistically and significantly different than the post-event proportions, and those being lower, then one could argue that the post-event record has a high probability of having been reworked. There is no way to know whether reworking or low contribution from "living" vegetation may be the case in the current manuscript. And, what is a "significantly higher proportion" of non-taeniate pollen near the base of the Wombarra Shale? How does that compare with the numbers below and above it?

The absence of glossopteris megafloral elements in, presumably, the core and elsewhere (unstated as to whether the correlative sections were assessed for macrofloral components) is used as evidence that glossopterid pollen in the upper palynozones must be reworked. What is the preservation potential of aerial leaves versus pollen/spores, comparatively? See comments above about how this statement can be tested. And, as the authors state further on, the interval between 778 and 587 m in the core is notable depauperate in macroremains. It's a core! What is the probability of dropping a coring device from altitude into a heavily forested area and actually intersect a tree? The low proportion, or poor preservation of macrofloral material in a core is not surprising. What is surprising is that the authors rely on this single drill core as part of their evidence. What about the presence of palynomorphs in the seven correlative sections illustrated in Figure 3? The authors do state that plant-fossil-bearing beds are present, in both outcrop and drill core, but no data are presented from any other locality than the latter (PHKB1). This is a shortcoming of the current work.

The authors state that the reader will find a diverse palynoflora of taeniate pollen with understorey components of ferns, sphenophytes, and lycophytes in Supplemental Table 3. Supplemental Table 3, though, doesn't break out these groups. Rather, the headings are: plant spores, pollen, % miospores (from which the reader, then, must do the calculation to determine the % megaspores [which appear to be an important part of the Wombarra Shale to lower Bulgo Sst interval's interpretation], a criterion used in the paper to interpret lycopsids), followed by other palynological categories. This reviewer presumes that the miospore category, in total, represents all three of the major plant clades, which would be the assumption based on what is provided. Maybe the wording needs to be changed in the text to better reflect an understorey of spore-producing groups because there is no breakdown of systematic affinities in the table.

Big claims made in this study, the loss of the glossopterid flora in eastern Australia, require big evidence. This reviewer finds that the evidence is obfuscated and not transparent. It is standard practice to count 300 palynomorphs to acquire a statistically reliable sample, and it is not clear that every sample analyzed consists of a 300-palynomorph count. Each sample consists of a 500 count in which acritarchs, dispersed organic matter, etc. are included. But, did the authors actually count 300 pollen/spores from each preparation. And, if not, are the spore/pollen data presented "raw" or "normalized" percentages to the requisite 300. There is no way for the reviewer to know what taxa have been identified, the proportion of glossopterid pollen versus all other non-glossopterid taxa, and how these proportions change stratigraphically. The claim of "demise" requires any future worker to be able to clearly see the trend that is claimed, and not just based on a generalized graphic. A spreadsheet of the raw data needs to be the supplemental resources, and a clear explanation of how these data were used in the study supplied to the reader.

The authors seem to have mixed up ecological terminology in their text. If elements of the *Glossopteris* flora have been documented by Dun (1908) in other parts of the basin, there can't be a "kill event" as proposed. The term used appears to be more "click bait" than real. What the authors might be able to claim is an extirpation of the glossopterid flora from this part of Australia, with a reduction in its biogeographic range (not killed off) to smaller areas in which the environmental requirements of the taxon were satisfied and populations continued in time. The question about a "brief" refugium, then, also plays into the query above about how the authors can distinguish between "reworked" glossopterid pollen and contribution from a living source without some statistical test of their data.

I understand the way in which the authors have registered their sampling horizons relative to the Bulli coal (0) and the distance below (in positive numbers) and above (in negative) numbers. Yet, this system is a bit confusing until the reader realizes that the negative stratigraphic position of a sample is above the horizon (it would take -565.35 m from the depth of 239.75 m to arrive at the top of the Bulli coal). Why not just use the depth in meters in the core, itself, without confusing the issue? The authors state that the FAD of *Lunatisporites* occurs at a depth of 745.6 m in the core which is -59.46 m from the top of the Bulli coal, which is ~ 60 m above their datum. Similarly, the authors refer to an abundance of pleuromeian megaspores at a depth of 614 m, which is -190 m from the top of the Bulli coal. Can't this be simplified because, currently, it is a bit convoluted? It is left to the reader to calculate what is meant by an "abundance" of megaspores from supplemental table 4. And, when this exercise is done, the proportion of megaspores at 614 m is 44%, which is more than the underlying sampling point (637 m @ 17%) but the two overlying sampling horizons (587, 559 m) may not be that statistically different (37% and 35%, respectively). From which lithofacies do these numbers originate? The addition of lithofacies and interpreted depositional environment to the supplemental tables would help the reader better understand the context of these assemblages.

Chemical Index of Alteration values, along with an increase in Nickel (did anyone attempt to evaluate for Hg or other heavy metals) are used to support the "loss" of the *Glossopteris* flora and turnover. Yet, if there are "refugia" in other parts of Australia and, potentially, the southern hemisphere paleocontinents, could this have only been a local effect rather than the presumed more global nature as implied? Could the "abrupt" collapse of the mire (and that infers that glossopterids were restricted to peat soils (histosols) when, in fact, they had a broader range of "wetland" habits) be just that. A change in physical conditions that prevented the accumulation of organic matter in peat swamps? The authors do state that there are "a few survivors" including *Glossopteris* elsewhere in the basin for a few meters above the boundary. Yet, a few meters above the mudrock in which these are preserved is an erosional contact (erosional phase of the landscape) with an overlying sandstone body. That diastem, alone, may be the culprit for the absence of this plant group in the area, as McManus et al. (2002) have reported the plant group from the early Triassic in Antarctica. Hence, this reviewer returns to the concept of extirpation rather than "kill."

Reviewers' comments:

Reviewer #1 (Remarks to the Author):

Fascinating, highly significant and much needed re-evaluation of the high latitude Permo-Triassic terrestrial record of Australia. Conclusions are highly significant – especially the lack of climate change across the extinction interval – contrasts with earlier claims from other regions. The precision dating also reveals the extinction pre-dates the marine extinction, as already inferred for lower latitude settings, again an important finding. The possibility of a diachronous *Glossopteris* extinction from low to high latitude is also potentially important.

We thank the reviewer for their praise of the significance of the manuscript.

I just have a few minor points:-

“Such a catastrophic and abrupt environmental change, if repeated, would have devastating consequences for life on Earth.” – self-evident therefore not needed, and bordering on pretentious.

The sentence has been rephrased to focus on the important parallels between the EPME and modern anthropogenic environmental change.

“well-calibrated Lopingian and Lower Triassic” I would replace ‘and’ with ‘to’
Corrected.

“Bulli seam in the south, Katoomba seam in the west, and Vales Point seam in the north of the basin” – shouldn’t seam be a capital ‘Seam’ as it is a proper noun?

We have corrected all these names to the current nomenclatural status given by the Australian Stratigraphic Names database.

Figure 2 – somewhere in the caption it needs to be stated that this a core record.

Updated.

“persistence of refugial glossopterid populations after the main ‘kill- event’ (into the P. crenulata Palynozone).” – delete “refugial” a loaded term implying migration to a safe area

which is inappropriate here, “holdover” is the correct term needed here.
Noted, and corrected.

As written the “Palaeofloras” section seem to record higher diversity above the extinction horizon than before it – see also figure 2 – would it be best to describe this as a turnover event rather than an extinction event?

This is an interesting observation, and something we have considered when writing this manuscript. A major extirpation of macrofossil elements of the classic Gondwanan *Glossopteris* flora is evident at the EPME. Numerous palynofloral taxa also go extinct within the extinction interval, but these are replaced in a stepwise manner by a series of incoming taxa with partly overlapping ranges. As a result, the diversity of the Early Triassic (and post-extinction event Permian) assemblages remains quite high throughout. As such, the terms ‘extinction event’ and ‘turnover’ are both relevant, and used in different parts of the manuscript in different contexts.

“not dissimilar to clay-pellet granulestones” – what’s the origin of these, if not reworking?

The clay-pellet breccias are indeed a product of local reworking within wetland and floodbasin environments. Our point, though, is that their thin, discontinuous, and rare occurrences indicate they did not represent a widespread, severe landscape degradation event, as postulated by Retallack (2005).

“The predominance of gray mudrocks immediately above the Bulli Coal and equivalents across the basin, together with sporadic phytoplankton cysts and AOM within these strata,” doesn’t this suggests marine influence, otherwise where do the phytoplankton come from? Was the Bulli Coal forest drowned/ transgressed?

We thank the reviewer for pointing out this. We have changed the text to to “The predominance of grey mudrocks immediately above the Bulli Coal and equivalents across the basin, together with sporadic phytoplankton and AOM within these strata, suggests that the extinction interval was characterized by generally moist conditions even after the loss of peat-forming floras.” The phytoplankton are not necessarily marine and we are gathering data from additional wells for a later paper dealing specifically with this issue. Therefore, we prefer to avoid implying any particular level of salinity in the current manuscript.

“by relatively short-ranging plant communities” – this needs clarifying because the next sentence then says that the next community appears 3 myr later, so the “short ranging” sounds like a very long time.

Thanks. This section has now been qualified, and written as follows: “...by a **succession of** relatively short-ranging plant communities...”. This indicates that each of these distinct short-ranging communities occurred one after the other. Furthermore, the 3 million year interval cited represents the time after the EPME at which corystosperm seed-ferns began to dominate assemblages. We have attempted to clarify the text to address this point and have added a literature reference.

Reviewer #2 (Remarks to the Author):

Fielding and co-authors present a multi-proxy study of the Bunnerong DDH-1 core from Sydney Basin. The core is about 1 km thick, and intersects the Permian-Triassic Boundary and mass extinction in a terrestrial sedimentary setting. Work on the core includes a sedimentologic log, palynology, major (CIA) and minor (Ni) element geochemistry, organic carbon isotopes, and the development of a single new U-Pb age via CA-ID-TIMS. I am not a paleontologist or paleobotanist, and so the most relevant aspect of my background is my experience with U-Pb geochronology. With regards to U-Pb, I see no potential issues. It comes from a top-shelf laboratory in BSU, and the data look great. The development of $\delta^{13}\text{C}_{\text{org}}$ records is also fairly standard, and cannot see any reason why the data is not reasonable.

But as a non-specialist, it is not clear to me how this work represents a major development in the study of the P-T mass extinction that would be appropriate for a generalist journal like Nature Communications. For example, the conclusions presented here appear no different from Metcalfe et al. (2015), which this MS cites and shares several co-authors with. In that contribution to Gondwana Research, 28 CA-ID-TIMS ages are presented from the Sydney and Bowen basins, as oppose to only one presented here. In their Fig. 15, Metcalfe et al. (2015) place the terrestrial extinction horizon right above the Bulli Coal, just like this MS; this figure also shows many of the same paleontologic/paleobiologic trends (loss of Glossopterids and coals, palynologic turnover) as that boundary that are discussed in this MS. Fig. 16 of Metcalfe et al. (2015) also summarizes $\delta^{13}\text{C}_{\text{org}}$ chemostratigraphies from across Australia, and they do not look substantially different from what is presented here.

We indicate in the text and in Fig. 2 that the P-Tr boundary is probably around 40 m lower in the succession than suggested by Metcalfe et al. (2015). Moreover we provide the first radiogenic isotope age constraints on the end-Permian mass extinction in Gondwana – showing it to be at the top of the uppermost coal seam (rather than at the top of the succeeding shale package) and somewhat earlier than the turnover in the marine record. Thus, we provide a major advance in unravelling the timing of the key events around the close of the Permian. Moreover, we show that this turnover in plant communities was contemporaneous with the onset of the primary extrusive phase of Siberian trap volcanism, which offers a causal linkage for this biotic disruption. Moreover, the paper by Metcalfe et al. (2015) lacks any details on quantitative palynology or the stratigraphic distribution of macrofossil groups within cores. In this paper, we synthesise ages from Metcalfe et al., (2015), the unpublished PhD thesis of Huyskens (2014), and our new data.

The single age developed here is a nice, new constraint to have, as it sits right during the start of the mass extinction event and thereby improves the precision offered of the extinction age. The CIA and Ni records are also new, and seem concordant with predictions about Earth system conditions predicted during the P-T extinction. The data should be published, but unless the authors can articulate more clearly why their work represents a significant advance that would appeal to a general audience, I think it should be submitted to a more specialist journal.

We reiterate:

- (1) That the multidisciplinary nature of the study is unprecedented for any analysis of the Permian-Triassic transition in continental settings;
- (2) That our sedimentological data indicate no major change in fluvial style across the EPME or PTB, and no abrupt aridification.
- (3) That our palaeontological data provide a biostratigraphic framework for evaluating changes in the palaeovegetation, and precisely identify the position of the extirpation of the *Glossopteris* flora (the EPME);
- (4) That our data provide a unique age control on the EPME, constraining this extinction event to some 410,000 years before the PTM and around 370,000 years before the main extinction in the marine realm;
- (5) That our collective data provide a basis for correlation to globally significant events (e.g stages within Siberian Trap volcanism) that may have been the primary drivers for the end-Permian mass extinction.

We have emended sections of the text to emphasize the novel data presented, and to highlight the significance of the findings with regard to global processes as noted by Reviewer 3.

A couple of comments:

-the climate modelling is barely discussed in this paper; the results seem entirely confined to the Results section, and do not come into the discussion really at all. I suggest removing it OR expanding it to become better integrated with the rest of the MS.

Climate modelling results have now been included in the Abstract, and a paragraph has been added to the discussion of the palaeoenvironmental interpretation, placing the model results in context with sedimentological and geochemical findings.

-in future versions, include line numbers on the text. It is very hard to provide specific comments otherwise. I have made some directly onto the PDF itself.

Thanks. Individual comments indicated on the pdf have now all been addressed.

Jon Husson

Reviewer #3 (Remarks to the Author):

The contribution by Fielding and colleagues focuses primarily on the Pacific Power Hawkesbury Bunnerong drill core, supplemented with a sedimentological context from correlative outcrop exposures, in which they present stratigraphic/sedimentologic, geochemic, palynologic, and geochronometric data, which are then used to model paleoatmospheric conditions in the latest Permian and earliest Triassic of eastern Australia. The authors proposed the onset of a short-lived climate perturbation approximately 370 ka prior to the onset of the end-Permian marine extinction event, wherein the demise (collapse) of the *Glossopteris* flora is proposed in this high paleolatitudinal position. This extirpation, rather than extinction (if other southern hemisphere records are accurate about the continuance of the flora in Antarctica), of the group in the Sydney Basin is used as a potential model for how vegetation might respond to climate-warming scenarios proposed for our current world.

What is exciting about the collective work of this research group is the multidisciplinary approach in a part of the world where much has been claimed by others. In many of the previous publications, few hard details have been placed into any high resolution lithostratigraphy constrained by geochronology, other than the work of Metcalfe and colleagues, whose interpretations about the placement of the end-Permian extinction and palynological biozones continue to be open to discussion. The fact that there is a notable loss of the glossopterid biome subsequent to the youngest peat swamp accumulation has been taken to represent a terrestrial reorganization as a consequence of some global perturbation.

The PHKB1 reference core is a little over 1000 m in length and, overall, is a prograding system from offshore marine shelf ending in “mostly” lowland alluvial plain settings. Peat (Bulli coal seam) accumulated in a coastal plain environment and palynological data from this interval shows a marked turnover in floral elements and “collapse” of the glossopterid flora. Euhedral zircons recovered from immediately above the coal, and analyzed by CA-ID-TIMS, yield a concordant date of 252.31 +/- 0.07 Ma, and these data are used to identify the continental End Permian Mass Extinction event.

This reviewer has several questions, most of which can be answered but one of which can't be addressed; that is, the “somewhat disturbed” nature of the core at the putative EPME. This latter concern will be addressed first. Nowhere in the text nor the supplemental materials is the “somewhat disturbed” nature described or presented. What, exactly, do the authors mean? This reviewer suspects that the coal (how much of its thickness, for example) had been removed from standard proximal analyses to assess its quality. And, if this horizon is identified as the point of turnover, is that based on the fact that the floristic elements in the peat are presumed to be glossopterid-dominated? Presumably, the Bulli coal is mined extensively elsewhere in the basin, and this reviewer wonders why a channel sample of that coal in a geographically close mine site wasn't sampled, processed, and used as a check on the core data. The overlying mudrock preserves marine acritarchs which indicate retrogradation in this part of the section; might a correlative site where coastal plain deposits that overlie the Bulli coal be appropriate to sample and assess, to determine whether there is such an “abrupt” loss that might not be a consequence of coastal plain inundation?

We indicate in the same sentence what is meant by “somewhat disturbed” – i.e the uppermost coal (Bulli Coal) is intruded by thin dolerite sills and much of it was removed for analysis at the time of drilling”. We have rephrased the sentence to clarify this. In any case, Figure 3 shows that the sedimentological pattern in this well is consistent with that represented in a series of wells forming a S–N transect across the basin.

The reviewer makes the assumption that the leiosphere acritarchs found in our study are ‘marine’. On the contrary, one of the primary findings of the present study is that the lithofacies indicate a consistent fluvial depositional system both immediately below and for a long interval above the End-Permian Mass Extinction. No indication of ‘marine’ transgression is evident. Further, there is a distinct absence of bioturbation, microbialites (as has been reported in some shallow marine P-T successions), hystrichosphaerids or other acritarchs typically found in Permian-Triassic marine strata (e.g., Schneebeli-Hermann et al., 2017), or other fossil or sedimentary evidence of marine conditions. We simply note that the presence

of phytoplankton indicates the persistence of moist conditions after the EPME, but in the absence of mire-forming deciduous glossopterid forests.

Given that glossopterids have long been interpreted as freshwater mire/lake-margin specialist plants, we argue that their absence (or extreme scarcity) in the beds overlying the Bulli Seam represents a genuinely abrupt disappearance from the landscape unrelated to moisture levels alone.

The authors identify a downcutting event above the Bulli coal in other parts of the basin, with the presence of a conglomeratic sandstone in erosional contact. They also use this evidence, in combination with geochemical data, to demonstrate a continuation of the same fluvial and pedogenic style across the landscape following peat termination.

The EPME occurs not at the base of the erosionally based sandstone/conglomerate but instead at the base of the shale below that sandstone. That shale does not have an erosional lower contact and appears to represent an interval of clastic deposition without any significant hiatuses. Our statement that “there was no major erosional event or catastrophic physical degradation of the landscape at the level of the continental EPME” remains valid. Fluvial erosion and downcutting occurred in a variety of locations in the aftermath of the EPME, but in a manner similar to that below the boundary. In many locations (see Figure 3), conversely, a thick fine-grained interval is preserved above the boundary. On a larger scale, there seems to have been no significant change in fluvial style through the uppermost Permian to lowermost Triassic succession apart from the termination of peat accumulation.

A second question arises about the palynological data, herein presented in a very generalized manner that is not wholly fulfilling nor transparent. The authors present pollen diagrams in which the proportion of spores vs. pollen are illustrated, along with aspects of phytoplankton and the dispersed organic matter (palynofacies; Fig. 2). Yet, the claim is that there is a collapse of the glossopterid flora without presenting the data to substantiate it. For example, why not provide the reader with plots of the percentage of glossopterid palynological components versus the percentage of pollen in each sample? Just plotting total spores vs. pollen tells the reader nothing about how the proportion of glossopterid representation has changed over the proposed “extinction” interval.

This aspect of the critique by Reviewer #3 relates to the recognition of glossopterid community collapse based on the fossil pollen signal. Whilst the palynology provides a clear signal of rapid turnover around the end of the Permian and into the Early Triassic, there is a universal problem with the use of the pollen record alone to illustrate a collapse in the flora. This is because (1) no single pollen taxon has been established as uniquely glossopterid. Numerous pollen types have been found within single glossopterid sporangia (Lindström et al., 1997); (2) some non-glossopterid seed fern groups produced morphologically equivalent taeniata bisaccate pollen; and (3) in a fluvial system, some degree of reworking of spores and pollen is inevitable so that last appearances are likely to extend beyond the demise of their parent plants. Our study benefits from a macrofossil analysis of the same core, wherein glossopterid leaves and roots, with unequivocal affinities and unlikely to be reworked, provide a more confident indicator of the extinction level by their termination at the top of the Bulli coal seam. Later in the text, we are more explicit about this, stating: “...but the absence of glossopterid macrofossils suggests that (taeniata bisaccate) grains were either reworked and/or produced by other gymnosperm groups.”

The presence of taeniate bisaccate pollen above the last occurrence of coals does not imply the persistence of glossopterids. In an attempt to clarify this issue, we have rephrased the text accordingly:

“The coal measures host a diverse palynoflora of which the pollen component is dominated by *Protohaploxylinus* and *Striatopodocarpidites*. Such taeniate (striate) bisaccate pollen were typically produced by glossopterids, although some grains of these morphotypes may derive from other seed fern groups. Nevertheless, we take the reviewer’s point that additional quantitative data could be expressed. On this basis, we have emended Fig. 2 to present quantitative data for six broad palynomorph groupings in order to highlight the localized perturbations of free-sporing and pollen-producing plants around the EPME and to indicate that otherwise the ratio of pteridophytic understorey spores to probably overstorey gymnosperm pollen remains relatively constant between the Permian and Early Triassic. Documentation of detailed species-level occurrences across multiple wells in the basin is indeed intended for a much longer paper in a specialist journal at a later date.

The authors note that they do continue to find “small” amounts of glossopteris pollen in overlying sediments, and attribute these (unspecified numbers) to having been reworked. As noted above, these (taeniate bisaccate) pollen grains do not signify the presence of glossopterids exclusively; furthermore, we are careful to point out in the text that “the absence of glossopterid macrofossils suggests that such [post-EPME] grains were either reworked and/or produced by other gymnosperm groups”.

The question of how much reworking of slightly older palynomorphs into slightly younger sediments is always a difficult problem to solve, mainly because there are few, if any, physical characteristics to separate these two populations. The reader is not informed about what percentage of glossopterid pollen is considered to represent such reworked palynomorphs. And, more importantly, the authors provide no statistical test to demonstrate the high probability of these “low numbers” as having been reworked, nor do they specify the interpreted depositional environment from which the palynomorphs were recovered. We have added a general comment to state that taeniate bisaccate pollen averages 4% of total palynomorphs through the first three palynozones above the EPME. We have emended the text to indicate that these low numbers of taeniate bisaccate grains extending into the Triassic occur in consistently fluvial facies (as evident from Figs 2 and 3).

How to test for “reworking” of glossopterid pollen? This reviewer suggests the possibility of using the proportion of glossopterid pollen recovered from siliciclastic deposits below and above the Bulli coal (but not from the coal, itself). They state that taeniate and non-taeniate bisaccate pollen average 3.9% and 4.4% in the lithologies above the Bulli coal. What are their proportions in siliciclastic lithologies below the Bulli coal?

We have emended Fig. 2 to show the proportions of these pollen types both above and below the EPME.

If the pre-event proportions are statistically different from the post-event proportions in the same depositional setting, controlling for potential taphonomic biases, then one could argue that the same proportion of glossopterid contribution outside of the peat swamps was occurring. In contrast, if the pre-event proportions are statistically and significantly different

than the post-event proportions, and those being lower, then one could argue that the post-event record has a high probability of having been reworked. There is no way to know whether reworking or low contribution from “living” vegetation may be the case in the current manuscript.

The approach outlined by the reviewer is an interesting angle for assessing the degree of reworking of pollen; however, in this specific case, the premise is that we can adequately identify glossopterid pollen (to the exclusion of similar pollen produced by other plants e.g. voltzialean conifers). As noted above, we cannot do that based on generalized morphology alone. In principle, this approach could be applied to any palynomorph group, but the problem would then be how to distinguish which grains are abundant as a by-product of reworking, and which are due to dominance in the contemporaneous ecosystem. This was undoubtedly a dynamic time in floral history, and discerning anomalous abundances due to reworking will be obfuscated by the (geologically) rapid fluctuations in both diversities and abundances of pollen taxa, even if we control for lithofacies/local depositional conditions. Just establishing a ‘baseline’ for comparison would be fraught with controversy.

Instead, we propose a qualitative, taphonomic approach to demonstrate the degree of reworking within this section; specifically, reworking of pre-EPME specimens into post-EPME strata. This has now been included in the supplementary materials section, but reiterated below:

Pre-EPME specimens typically have a high degree of corrosion. The style of degradation is characteristic of authigenic sulphide mineralisation (e.g., pyrite, marcasite). As authigenic sulphides have been shown to form within organic-rich reducing conditions during or soon after sedimentation, this diagenetic effect on pre-EPME specimens was likely (pene)contemporaneous. Thus, this degradation pattern serves as a proxy for specimens that have been reworked from pre-EPME sediments into overlying strata. This effect is not likely a local phenomenon as it has been noted in other wells from the Sydney Basin (C. Mays, pers. obs.). However, the occurrence of such degraded grains in post-EPME strata is negligible, indicating only minor reworking. We present the photographic evidence in Supplemental Figure 3 to support this interpretation.

And, what is a “significantly higher proportion” of non-taeniate pollen near the base of the Wombarra Shale? How does that compare with the numbers below and above it?

These details were included in the following paragraph of the manuscript, but we notice that there was some redundancy in the text. As such, we have trimmed the text, and brought the relevant statistics of taeniate vs non-taeniate grains into the preceding paragraph. Further we have upgraded Fig. 2 to include this data for the whole succession.

The absence of glossopteris megafloral elements in, presumably, the core and elsewhere (unstated as to whether the correlative sections were assessed for macrofloral components) is used as evidence that glossopterid pollen in the upper palynozones must be reworked.

We raise the possibility of minor reworking, but we also state within the manuscript that some glossopterid-like taeniate bisaccate pollen were probably also produced by non-glossopterid plants. This has now been clarified in the text, and discussed above in our response to the reviewer.

What is the preservation potential of aerial leaves versus pollen/spores, comparatively?

Since spores and pollen are produced in vastly greater numbers than leaves, the former will always have a greater representation than the latter in non-oxidized sediments. Robyn Burnham and colleagues published a series of modern analogue studies demonstrating that fossil leaves deposited in fluvial overbanks (the probable depositional conditions of the macrofloral assemblages of PHKB1) strongly reflect the local floral assemblages (Burnham, 1989, 1994; Burnham et al., 1992). As such, these fossils are likely not subject to reworking, and thus ought to be representative of the contemporaneous floras. In regards to the predisposition of spore-pollen reworking, this has long been recognised as an important factor in Quaternary palynological studies (e.g., Stanley, 1966; Campbell, 1999), and can even occur over long geological intervals (Legault and Norris, 1982). However, our palynological samples were consistently collected from coals, carbonaceous siltstone and v-fine sandstone facies; pollen deposited within these facies have been shown to have the lowest proportion of reworked specimens (Fall, 1987; Campbell, 1999). Furthermore, as we have demonstrated above, there is only minimal evidence of long-term reworking (e.g., pre-EPME specimens into post-EPME strata) within the PHKB1 region.

See comments above about how this statement can be tested. And, as the authors state further on, the interval between 778 and 587 m in the core is notable depauperate in macroremains. It's a core! What is the probability of dropping a coring device from altitude into a heavily forested area and actually intersect a tree?

The chances are actually quite high since most deciduous forest communities have a rich leaf-litter layer.

The low proportion, or poor preservation of macrofloral material in a core is not surprising. What is surprising is that the authors rely on this single drill core as part of their evidence. What about the presence of palynomorphs in the seven correlative sections illustrated in Figure 3? The authors do state that plant-fossil-bearing beds are present, in both outcrop and drill core, but no data are presented from any other locality than the latter (PHKB1). This is a shortcoming of the current work.

Because of the multi-proxy nature of the study, and because of manuscript size limitations, we have focused on this single well in the present manuscript in order to establish a reference section for later studies. The sedimentary, palynological, palaeobotanical and geochemical signals from this reference well are intended to facilitate future comparisons with additional wells from across the Sydney Basin. These data are presently being collected, and will be prepared for a series of future publications of a more specialist nature.

The authors state that the reader will find a diverse palynoflora of taeniate pollen with understorey components of ferns, sphenophytes, and lycophytes in Supplemental Table 3. Supplemental Table 3, though, doesn't break out these groups.

This was a mistake, and has now been corrected in the text.

Rather, the headings are: plant spores, pollen, % miospores (from which the reader, then, must do the calculation to determine the % megaspores [which appear to be an important part of the Wombarra Shale to lower Bulgo Sst interval's interpretation], a criterion used in the paper to interpret lycopsids), followed by other palynological categories. This reviewer presumes that the miospore category, in total, represents all three of the major plant clades, which would be the assumption based on what is provided. Maybe the wording needs to be

changed in the text to better reflect an understory of spore-producing groups because there is no breakdown of systematic affinities in the table.

Because of manuscript size limitations, only the most relevant details of a broader quantitative dataset have been presented herein – principally to establish a biozonation for the succession and to aid palaeoenvironmental interpretations. As mentioned above, the detailed palynomorph count data are presently being compiled across multiple wells for a future publication. Megaspores are recorded in the mesofossil fraction of macerated samples and are not included in the miospore counts. A general statement on megaspore abundance is now included in the text.

Big claims made in this study, the loss of the glossopterid flora in eastern Australia, require big evidence. This reviewer finds that the evidence is obfuscated and not transparent. It is standard practice to count 300 palynomorphs to acquire a statistically reliable sample, and it is not clear that every sample analyzed consists of a 300-palynomorph count. Each sample consists of a 500 count in which acritarchs, dispersed organic matter, etc. are included. But, did the authors actually count 300 pollen/spores from each preparation. And, if not, are the spore/pollen data presented “raw” or “normalized” percentages to the requisite 300.

In addition to the palynofacies counts of 500 (from kerogen slides), broad palynomorph groups and index taxa were identified from counts of 250 per sample. This was indicated in the ‘Methods’ section, but this has now been clarified. Note: the total palynomorph count was 250 + additional palynomorphs from the palynofacies counts. Palynomorph counts of ≥ 250 were chosen to ensure valid comparisons with count sizes in a recent study of the End-Permian from a high northern palaeolatitude locality (Greenland; Schneebeli-Hermann et al., 2017).

There is no way for the reviewer to know what taxa have been identified, the proportion of glossopterid pollen versus all other non-glossopterid taxa, and how these proportions change stratigraphically. The claim of “demise” requires any future worker to be able to clearly see the trend that is claimed, and not just based on a generalized graphic. A spreadsheet of the raw data needs to be in the supplemental resources, and a clear explanation of how these data were used in the study supplied to the reader.

These points have been addressed above; namely, glossopterid pollen are not considered to be consistently distinguishable from some other groups herein, and the full palynomorph count data from this and associated wells are being compiled for a future palynology-focused manuscript. We provide quantitative data for an expanded range of spore-pollen groups in the Supplementary Materials. We again emphasize that the macrofossil record provides the more secure signal of the disappearance of glossopterids from the succession.

The authors seem to have mixed up ecological terminology in their text. If elements of the *Glossopteris* flora have been documented by Dun (1908) in other parts of the basin, there can't be a “kill event” as proposed. The term used appears to be more “click bait” than real. What the authors might be able to claim is an extirpation of the glossopterid flora from this part of Australia, with a reduction in its biogeographic range (not killed off) to smaller areas in which the environmental requirements of the taxon were satisfied and populations continued in time.

The Early Triassic record of *Glossopteris* from Antarctica by McManus et al. (2002) now appears to be equivocal. In the Sydney Basin, the very last putative occurrences of

glossopterid macrofossils occur in shales below (not at) the locally erosional contact with the Coal Cliff Sandstone and equivalent units. At most, a few representatives of the glossopterids may have persisted briefly after the EPME, but we have no evidence of them continuing into the Triassic proper. We have rephrased portions of the text to account for these points.

The question about a “brief” refugium, then, also plays into the query above about how the authors can distinguish between “reworked” glossopterid pollen and contribution from a living source without some statistical test of their data.

Evidence for pollen reworking is addressed above. Our suggestion of short-lived and very minor holdover populations is based on published records of macrofossils (leaves and roots) alone.

I understand the way in which the authors have registered their sampling horizons relative to the Bulli coal (0) and the distance below (in positive numbers) and above (in negative) numbers. Yet, this system is a bit confusing until the reader realizes that the negative stratigraphic position of a sample is above the horizon (it would take -565.35 m from the depth of 239.75 m to arrive at the top of the Bulli coal). Why not just use the depth in meters in the core, itself, without confusing the issue? The authors state that the FAD of Lunatisporites occurs at a depth of 745.6 m in the core which is -59.46 m from the top of the Bulli coal, which is ~ 60 m above their datum. Similarly, the authors refer to an abundance of pleuromeian megaspores at a depth of 614 m, which is -190 m from the top of the Bulli coal. Can't this be simplified because, currently, it is a bit convoluted? It is left to the reader to calculate what is meant by an “abundance” of megaspores from supplemental table 4. And, when this exercise is done, the proportion of megaspores at 614 m is 44%, which is more than the underlying sampling point (637 m @ 17%) but the two overlying sampling horizons (587, 559 m) may not be that statistically different (37% and 35%, respectively). From which lithofacies do these numbers originate? The addition of lithofacies and interpreted depositional environment to the supplemental tables would help the reader better understand the context of these assemblages.

We have emended the system for registering sampling horizons in Supplemental Table 4 for clarity. We provide only a general statement on the abundance of megaspores, since these do not occur in the sieved fraction containing miospores. These were recovered from a separate (mesofossil-sized) fraction of the samples and can only be documented in terms of their relative abundance ranging from absent to >50 per 60 g sample. This is now clarified in the text.

Chemical Index of Alteration values, along with an increase in Nickel (did anyone attempt to evaluate for Hg or other heavy metals) are used to support the “loss” of the *Glossopteris* flora and turnover. Yet, if there are “refugia” in other parts of Australia and, potentially, the southern hemisphere paleocontinents, could this have only been a local effect rather than the presumed more global nature as implied? Could the “abrupt” collapse of the mire (and that infers that glossopterids were restricted to peat soils (histosols) when, in fact, they had a broader range of “wetland” habits) be just that. A change in physical conditions that prevented the accumulation of organic matter in peat swamps? The authors do state that there are “a few survivors” including *Glossopteris* elsewhere in the basin for a few meters above the boundary. Yet, a few meters above the mudrock in which these are preserved is an

erosional contact (erosional phase of the landscape) with an overlying sandstone body. That diastem, alone, may be the culprit for the absence of this plant group in the area, as McManus et al. (2002) have reported the plant group from the early Triassic in Antarctica. Hence, this reviewer returns to the concept of extirpation rather than “kill.”

The analytical methodologies used in this study did not allow for high-resolution determinations of Hg content. Regardless, there are some significant issues associated with using Hg as a proxy for volcanism. The first issue relates to the numerous potential sources of Hg, including the combustion of coal deposits, biomass burning, and soil erosion (Berquist, 2017, PNAS). Because Hg is sensitive to anoxia, it may undergo transformations and possible migration. Distinguishing between these potential sources and mechanisms of Hg accumulation requires the use of Hg isotopes, which is beyond the scope of the present study. Another concern relates to local volcanism in the Sydney Basin region, which could have contributed Hg to the depositional system. By contrast, there are no local sources of Ni in the Sydney Basin, making it a preferred proxy for this particular study. We do hope to analyze Hg in future studies, but this measurement is currently beyond the scope of the current work.

LIST OF MAIN REVISIONS TO THE MANUSCRIPT

1. The abstract has been completely revised to highlight additional key findings.
2. Corrected lithostratigraphic nomenclature.
3. Emendation of Fig 1 to make locality names compatible with Fig. 3, and addition of a legend to Figure 3.
4. Addition of further quantitative data to the palynological component of Fig. 2 (and Supplementary Table 2) to segregate the proportional representation of additional palynomorph groups through time.
5. Clarification of methodological details of palynological data collection.
6. Additional text and figure to supplementary materials to assess the degree of palynological reworking.
7. Clarification of the criteria for placement of the End-Permian Mass Extinction.
8. Updated data on the patterns of palynological turnover from the latest Permian to Early Triassic.
9. Clarification of the depositional environment of beds deposited immediately after the EPME.
10. Presentation of the evidence for reworked pre-EPME palynomorphs and compilation of an additional supplemental figure for this purpose.
11. Clarification of the affinities of taeniate bisaccate pollen.
12. Comments annotated on the manuscript pdf by Reviewer 3 have been addressed.
13. Emended the system for registering sampling horizons in Supplemental Table 2
14. Text has been added to integrate the climate modelling with the palaeoenvironmental interpretation.
15. Aspects of the sedimentology and geochemistry have been clarified, as requested.

Additional references mentioned in the response to the reviewers' comments:

- Burnham, R.J., 1989. Relationships between standing vegetation and leaf litter in a paratropical forest: Implications for paleobotany. *Review of Palaeobotany and Palynology* 58, 5–32.
- Burnham, R.J., 1994. Patterns in tropical leaf litter and implications for angiosperm paleobotany. *Review of Palaeobotany and Palynology* 81, 99–113.
- Burnham, R.J., Wing, S.L., Parker, G.G., 1992. The reflection of deciduous forest communities in leaf litter: implications for autochthonous litter assemblages from the fossil record. *Paleobiology* 18, 30–49.
- Campbell, I.D., 1999. Quaternary pollen taphonomy: examples of differential redeposition and differential preservation. *Palaeogeography Palaeoclimatology Palaeoecology* 149, 245–256.
- Fall, P.L., 1987. Pollen taphonomy in a canyon stream. *Quaternary Research* 28, 393–406.
- Goldhaber, M.B., 2007. Part 7.10 Sulfur-rich sediments. In: Holland, H.D., Turekian, K.K. (Eds), *Treatise on Geochemistry* 7. Elsevier Ltd., pp. 257–288.
- Legault, J.A., Norris, G., 1982. Palynological evidence for recycling of Upper Devonian into Lower Cretaceous of the Moose River Basin, James Bay Lowland, Ontario. *Canadian Journal of Earth Sciences* 19, 1–7.
- Lindström, S., McLoughlin, S., Drinnan, A.N., 1997. Intraspecific variation of taeniate bisaccate pollen within Permian glossopterid sporangia, from the Prince Charles Mountains, Antarctica. *International Journal of Plant Sciences* 158, 673–684.
- Neves, R., Sullivan, H.J., 1964. Modification of fossil spore exines associated with the presence of pyrite crystals. *Micropaleontology* 10, 443–452.
- Schneebeil-Hermann, E., Hochuli, P.A., Bucher, H., 2017. Palynofloral associations before and after the Permian–Triassic mass extinction, Kap Stosch, East Greenland. *Global and Planetary Change* 155, 178–195.
- Stanley, E.A., 1966. The problem of reworked pollen and spores in marine sediments. *Marine Geology* 4, 397–408.

Reviewers' Comments:

Reviewer #3:

Remarks to the Author:

The authors have provided convincing evidence in this amplified revised version of their analysis of PHKB1 drill core in the Sydney Basin, Australia, in which they have identified a short, 7-m stratigraphic interval over which elements of the Glossopteris flora are shown to meet their local demise. Using geochronometric and geochemical evidence, the authors propose that the loss of this wetland biome was a consequence of a short, rapid warming coincident with the onset of Siberian Trap eruptions. They demonstrate that neither an aridification trend nor a change in fluvial regime accompanied the transition across the upper Permian to lower Triassic succession, as reported, and intimated, elsewhere.

I believe that the authors have made an honest attempt at bringing transparency to their palynological data which, in the previous version, was a bit obtuse. I'm unsure about why another reviewer had a problem with the presence of phytoplankton in the fluvial deposits overlying the Bulli coal. The introduction of marine phytoplankton into coastal river systems, transporting them up to 100 km from the shoreline, is a feature of regimes influenced by meso- and macrotidal processes (see: Staub et al., 2000).

The one sticking point with this reviewer is the choice of the term "mass extinction" rather than, possibly, "selective extinction" in the flora. In effect, only members of the wetland assemblage and, in particular, the glossopterids are affected. And, although the authors now acknowledge that their loss in the Sydney Basin may be a regional extirpation (ln. 217; and this term also would be acceptable in the title), this potential fact (which needs to be substantiated elsewhere in a geochronometric context) seems to be overlooked in preference of the "mass extinction" interpretation. If, indeed, there are "holdovers" in other parts of the paleoSouthern Hemisphere, then there can't be a "mass extinction" by definition. Similarly, the authors note that (lns. 376-378) "the plant macrofossil and palynological records indicate major shifts in plant-group representation through this interval." Major shifts and the loss of one clade doesn't equate to a "mass" extinction event. Changing one word in the title resolves these issues.

Nevertheless, I urge the Editors to accept this contribution because it is very important both in the data presented and in the coincidental timing of the biotic response to an interpreted warming trend. I agree with the authors that this research provides "a major advance in unraveling the timing of key events around the end of the Permian." It is relevant to today's conditions, and is another model (albeit there are very few data-driven models at the moment) as to how terrestrial ecosystems responded to perturbation in the latest Permian.

Summary of changes

The comment by referee 3 criticising use of the term “mass extinction”:

The one sticking point with this reviewer is the choice of the term “mass extinction” rather than, possibly, “selective extinction” in the flora. In effect, only members of the wetland assemblage and, in particular, the glossopterids are affected. And, although the authors now acknowledge that their loss in the Sydney Basin may be a regional extirpation (ln. 217; and this term also would be acceptable in the title), this potential fact (which needs to be substantiated elsewhere in a geochronometric context) seems to be overlooked in preference of the “mass extinction” interpretation. If, indeed, there are “holdovers” in other parts of the paleoSouthern Hemisphere, then there can’t be a “mass extinction” by definition. Similarly, the authors note that (lns. 376-378) “the plant macrofossil and palynological records indicate major shifts in plant-group representation through this interval.” Major shifts and the loss of one clade doesn’t equate to a “mass” extinction event. Changing one word in the title resolves these issues.

has been addressed by changing the expression “EPME” (for end-Permian Mass Extinction) to “EPE” (for end-Permian Extinction) in the title, throughout the paper, in Figure 2, and in the Supplementary document.